# An Analytical Theory of Curriculum Learning in Teacher-Student Networks

**Luca Saglietti[†,∗], Stefano Sarao Mannelli[‡,∗], and Andrew Saxe[‡,§]**

## Abstract

In animals and humans, curriculum learning—presenting data in a curated order—is critical to rapid learning and effective pedagogy. A long history of experiments has demonstrated the impact of curricula in a variety of animals but, despite its ubiquitous presence, a theoretical understanding of the phenomenon is still lacking. Surprisingly, in contrast to animal learning, curricula strategies are not widely used in machine learning and recent simulation studies reach the conclusion that curricula are moderately effective or even ineffective in most cases. This stark difference in the importance of curriculum raises a fundamental theoretical question: when and why does curriculum learning help? In this work, we analyse a prototypical neural network model of curriculum learning in the high-dimensional limit, employing statistical physics methods. We study a task in which a sparse set of informative features are embedded amidst a large set of noisy features. We analytically derive average learning trajectories for simple neural networks on this task, which establish a clear speed benefit for curriculum learning in the online setting. However, when training experiences can be stored and replayed the advantage of curriculum in standard neural networks disappears, in line with observations from the deep learning literature. Inspired by synaptic consolidation techniques developed to combat catastrophic forgetting, we propose curriculum-aware algorithms that consolidate synapses at curriculum change points and investigate whether this can boost the benefits of curricula. We derive generalisation performance as a function of consolidation strength (implemented as an $L_2$ regularisation/elastic coupling connecting learning phases), and show that curriculum-aware algorithms can yield a large improvement in test performance. Our reduced analytical descriptions help reconcile apparently conflicting empirical results, trace regimes where curriculum learning yields the largest gains, and provide experimentally-accessible predictions for the impact of task parameters on curriculum benefits. More broadly, our results suggest that fully exploiting a curriculum may require explicit adjustments in the loss.

## 1 Introduction

Presenting learning materials in a meaningful order according to a curriculum greatly helps learning in animals and humans [1, 2, 3, 4], and is considered an essential aspect of good pedagogy [5]. For example, humans have been shown to learn visual discriminations faster when presented with examples that exaggerate the relevant difference between classes, a phenomenon known as "fading" [6, 7, 8]. Beyond humans, curricula in the form of "shaping" or "staircase" procedures are a near-universal feature of task designs in animal studies, without which training often fails entirely. For

† Department of Computing Sciences, Bocconi University.
‡ Gatsby Computational Neuroscience Unit & Sainsbury Wellcome Centre, University College London.
§ FAIR, Meta AI
∗ Equal contributions.

36th Conference on Neural Information Processing Systems (NeurIPS 2022).

instance, the International Brain Laboratory task, a standardised perceptual decision-making training paradigm in mice, involves six stages of increasing difficulty before reaching final performance [9].

Building from this intuition, a seminal series of papers proposed a similar curriculum learning approach for machine learning (ML) [10, 11, 12]. In striking contrast to the clear benefits of curriculum in biological systems, however, curriculum learning has generally yielded equivocal benefits in artificial systems. Experiments in a variety of domains [13, 14] have found usually modest speed and generalisation improvements from curricula. Recent extensive empirical analyses have found minimal benefits on standard datasets [15]. Indeed, a common intuition in deep learning practice holds that training distributions should ideally be as close as possible to testing distributions, a notion which runs counter to curriculum. Perhaps the only areas where curricula are actively used are in large language models [16] and certain reinforcement learning settings [17].

This gap between the effect of curriculum in biological and artificial learning systems poses a puzzle for theory. When and why is curriculum learning useful? What properties of a task determine the extent of possible benefits? What ordering of learning material is most beneficial? And can new learning algorithms better exploit curricula? Compared to the empirical investigations of curriculum learning, theoretical results on curriculum learning remain sparse. Most notably, [18, 19] show that curriculum can lead to faster learning in a simple setting, but the effects of curriculum on asymptotic generalisation and the dependence on task structure remain unclear. A hint that indeed curriculum learning might lead to statistically different minima comes from a connection between constraint-satisfaction problems and physics results on flow networks [20], but to our knowledge no direct result has been reported in the modern theoretical ML literature.

In this work we study the impact of curriculum using the analytically tractable teacher-student framework and the tools of statistical physics [21, 22, 23, 24]. High-dimensional teacher-student models are a popular approach for systematically studying learning behaviour in neural networks [25, 26, 22], and have recently been leveraged to analyse a variety of phenomena [27, 28, 29, 30, 31, 32]. Using a simple model to build structured data [12], we examine the impact of ordering examples by increasing difficulty (curriculum), decreasing difficulty (anti-curriculum), or standard shuffled training. We derive exact expressions for the online learning dynamics and the performance of batch learning. However, in the latter, curriculum confers no benefit under standard training in our model setting. Motivated by theories of synaptic consolidation and elastic weight consolidation [33, 34], we introduce elastic penalties (Gaussian priors) that regularise training toward solutions obtained in prior curriculum phases, instantiating a long-term memory effect. With these priors, curriculum yields benefits both in the online 3 and in the batch 4 settings.

**Further related work.** The first empirical investigation of curriculum learning appeared in 1927 [35], consisting in a visual discrimination task for dogs under curriculum and no-curriculum paradigms. Later behavioural studies proved curricula to be beneficial independent of the animal (dogs, mice, rats, pigeons, humans) and the data modality (visual, auditory, or tactile stimuli) [36, 1, 2, 37, 38, 6]. However, these experimental observations were not observed in standard artificial neural networks (ANNs). Several ideas in the connectionist community were proposed in order to show curriculum effects in the learning dynamics of ANNs [39, 40, 10, 11]. While these studies were able to match previous experimental data, they also required substantial changes in the architecture of the ANN and/or in the learning rule.

Except for very few instances [16, 17], standard ML practice tends to avoid taking curricula into account. An obvious obstacle is the fact that most datasets do not provide meta-data about sample difficulties. An interesting line of research pointed out the possible relevance of implicit curricula, based on the observation that neural networks tend to consistently learn the samples in a certain order [41]. Thus, a possible way of addressing the lack of difficulty labels would be to use the natural learning order as indicative of the various difficulties of the training samples. However, a recent work [15], which compared several heuristics for curriculum learning —including implicit curricula— in a variety of settings, showed limited benefits with this strategy.

The picture that emerges from the literature seems contradictory: on the one hand, curricula appear fundamental to biological learning; on the other hand, curricula appear largely irrelevant in many machine learning settings. The core motivation behind our work is to reconcile these views and contribute to a theoretical understanding of curriculum learning.

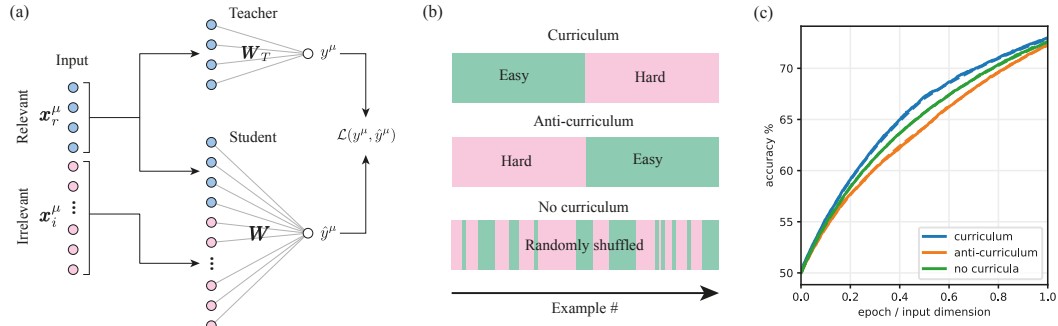

Figure 1: **Teacher-student setting for curriculum learning.** (a) Illustration of teacher-student setting in which a "student" network is trained from *i.i.d.* inputs with labels from a "teacher" network. Since the teacher network is sparse, its output depends only on a subset of *relevant* input features. (b) We consider curricula which order examples by difficulty, here taken to be the variance in the irrelevant feature dimensions. We refer to increasing, decreasing, and random difficulty order as curriculum, anti-curriculum, and no curriculum, respectively. (c) Example test error on hard examples for the student over training. The switch-point between easy and hard samples lies at $\alpha = 1/2$. Solid lines show numerical simulations, while dashed lines show theoretical predictions derived in Section 3. For this particular parameter setting, curriculum speeds learning but only modestly improves final performance at $\alpha = 1$. Parameters: $\alpha_1 = 1$, $\alpha_2 = 1$, $\Delta_1 = 0$, $\Delta_2 = 1$, $\gamma = 10^{-5}$, $\eta = 3$.

## 2 Model definition and overview of approach

In the following, we revisit a prototypical model of curriculum learning from [12] that finds correspondence to the fading literature [6] as highlighted in Sec. 5. Our setting is summarised in Fig. 1. The model entails a simple teacher-student setup, where teacher and student are each shallow 1-layer neural networks of size $N$ (also known as perceptrons). The learning task for the student is a binary classification problem, with dataset $\mathcal{D} = \{(y^\mu, \boldsymbol{x}^\mu)\}_{\mu=1}^M$, where the ground-truth labels are produced by the teacher network $y^\mu = \text{sign } \boldsymbol{W}_T \cdot \boldsymbol{x}^\mu$. The student learns via empirical risk minimisation of an $L_2$ regularised convex loss.

A key feature of this model is that the teacher network is sparse, with only a fraction $\rho < 1$ of $\sim \mathcal{N}(0,1)$ non-zero components. Therefore, in order to achieve a good test accuracy, the student has to guess which components should be set to zero and align the relevant weights in the correct direction. A large range of $0 < \rho < 1$ could give rise to the phenomenology we seek to analyse. In the remainder of the paper we will focus on the case $\rho = 0.5$.

We model the variable degree of difficulty in the samples by decomposing each input vector as $\boldsymbol{x}^\mu = [\boldsymbol{x}_r^\mu, \boldsymbol{x}_i^\mu] \in \mathbb{R}^N$, where $\boldsymbol{x}_r^\mu \in \mathbb{R}^{\rho N}$ denotes the relevant components of the input, and $\boldsymbol{x}_i^\mu \in \mathbb{R}^{(1-\rho)N}$ the irrelevant ones. Note that, crucially, the sparse teacher network is completely blind to the irrelevant part of the input: $y^\mu = \text{sign } \sum_{j=1}^{\rho N} W_{T,j} x_{r,j}^\mu$. While $x_{r,j}^\mu$ i.i.d. $\mathcal{N}(0,1), \forall \mu,$[1] we consider the variance for the irrelevant components to be sample-dependent $x_{i,j}^\mu \sim \mathcal{N}(0, \Delta^\mu)$. A smaller variance in the irrelevant part induces a higher SNR in the student learning problem.

The dataset is partitioned according to difficulty levels given by the variances of the irrelevant inputs. For simplicity we consider only two partitions in most of our analysis, but generalisations to multiple difficulty levels follow straightforwardly. We thus have a dataset with $M = (\alpha_1 + \alpha_2)N = \alpha N$ samples in total. In the first $\alpha_1 N$ samples the irrelevant inputs have variance $\Delta_1$, while for the remaining $\alpha_2 N$ samples the variance is $\Delta_2 > \Delta_1$. In the curriculum learning condition we present the easy examples first, while in the anti-curriculum condition we present the hard examples first. Standard learning presents examples shuffled in random order.

---

[1]In [12] the input distribution is uniform between 0 and 1, but this does not qualitatively change the results.

# 3  Online dynamical solution in the large input limit

We start by focusing on the same online learning setting explored in [12]. We consider a 1-layer student network with sigmoidal activation function, $\sigma(\cdot) = \text{erf}(\cdot/\sqrt{2})$, that learns to minimise a mean square error loss with $L_2$ regularisation of intensity $\gamma$, using gradient descent. This yields the updates

$$\boldsymbol{W}^{\mu+1} = \boldsymbol{W}^\mu - \frac{\eta}{\sqrt{N}} \sigma'\left(\frac{\boldsymbol{W}^\mu \cdot \boldsymbol{x}^\mu}{\sqrt{N}}\right)\left(\sigma\left(\frac{\boldsymbol{W}^\mu \cdot \boldsymbol{x}^\mu}{\sqrt{N}}\right) - y^\mu\right)\boldsymbol{x}^\mu - \gamma\boldsymbol{W}^\mu. \tag{1}$$

The dynamics of the model can be analysed in the high-dimensional limit $N, M \to \infty$ with $\alpha = M/N = \mathcal{O}(1)$. Generalising the results of [26, 42] on the online stochastic gradient descent dynamics in single-layer regression problems, we obtain a precise description of the performance at all times, as a function of several order parameters: the squared norm of the relevant and irrelevant part of the student weights $Q_r = \frac{1}{N}\boldsymbol{W}^r \cdot \boldsymbol{W}^r$ and $Q_i = \frac{1}{N}\boldsymbol{W}^i \cdot \boldsymbol{W}^i$, respectively; the overlap of the relevant weights of the student and teacher $R = \frac{1}{N}\boldsymbol{W}^r \cdot \boldsymbol{W}_T$; and the squared norm of the teacher vector $T = \frac{1}{N}\boldsymbol{W}_T \cdot \boldsymbol{W}_T$. In particular, given $Q_r$, $Q_i$, $R$ and $T$, the test loss (i.e. average loss on a new example) on a dataset with variance $\Delta$ in the irrelevant inputs is given by

$$\mathcal{L}_{\text{MSE}} = \frac{1}{2} + \frac{1}{\pi}\sin^{-1}\frac{Q_r + \Delta Q_i}{1 + Q_r + \Delta Q_i} - \frac{2}{\pi}\sin^{-1}\frac{R/\sqrt{T}}{\sqrt{Q_r + \Delta Q_i + 1}},$$

the accuracy by

$$\mathcal{A} = \mathbb{E}\left[\frac{1}{2}(y\,\text{sign}\hat{y} + 1)\right] = \frac{1}{2} + \frac{1}{\pi}\sin^{-1}\left(\frac{R}{\sqrt{T(Q_r + \Delta Q_i)}}\right). \tag{2}$$

If the dataset contains a random mixture of different difficulty levels $\Delta_1, \Delta_2, \ldots$, the loss and accuracy can be obtained by taking a weighted average over the partitions.

To understand how test performance changes through learning, we study the evolution of the order parameters. Combining their definition with the definition of the dynamics (1) and the fact that the random variables concentrate in the high-dimension as $N \to \infty$, we obtain an analytic form for the updates: $Q_r \leftarrow f_{Q_r}(Q_r, Q_i, R, T), Q_i \leftarrow f_{Q_i}(Q_r, Q_i, R, T), R \leftarrow f_R(Q_r, Q_i, R, T)$; where $f_{Q_r}$, $f_{Q_i}$ and $f_R$ are long but explicit expressions that are reported in the supplementary material (SM).

**Dynamical advantages of curriculum.**   With these theoretical results in hand, we can now characterise the performance of curricula in the online setting. We obtain a description of the learning trajectories for each learning protocol, yielding the evolution of training and test accuracies, and of other observables such as the norm of the student and its overlap with the teacher.

Solving the dynamical equations gives two key advantages relative to simulating models in this setting. First, they are free of finite size effects and stochastic fluctuations. And second, their evaluation is very fast (up to 6 orders of magnitude in simulation time reduction see SM **??**), enabling systematic exploration of the parameter space of the problem, along with fine-grained optimisation over hyper-parameters such as learning rate, weight decay and scaling in the initialisation.

Optimising final test accuracy separately for each curriculum strategy, we find that curriculum learning is the optimal strategy, followed by baseline (no-curriculum) and lastly anti-curriculum. In Fig. 1c we show typical learning trajectories for a dataset with equal numbers of easy and hard samples. The results of the simulations (solid lines) are well-described by our theoretical equations (dashed lines), and show that the curriculum strategy leads to better performance throughout training. Fig. 1c shows the evolution during training of the test accuracy computed on the whole dataset.

Next, we systematically trace the effect of curriculum for a range of total dataset sizes $(\alpha_1 + \alpha_2)$ and number of easy examples $\alpha_1$ in the phase diagram in Fig. 2. This diagram shows in panels (a) and (b) the accuracies on hard instances reached at the end of training, by curriculum learning and anti-curriculum learning respectively, normalised by the accuracy reached by the standard strategy. The two heatmaps show that curriculum learning always outperforms standard learning and that, on the other hand, anti-curriculum learning outperforms standard learning only in part of the diagram. Comparing the two strategies, in Fig. 2 (c), we can observe that there is a region for small $\alpha$ and $\alpha_1$ where anti-curriculum learning is the best strategy, while in the majority of the situations curriculum

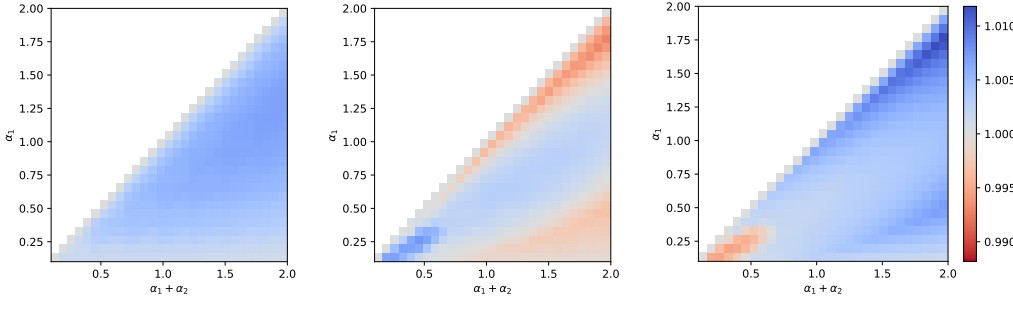

(a) Curriculum learning.     (b) Anti-curriculum learning.     (c) Curriculum vs anti-curriculum.

Figure 2: **Phase diagram of online learning performance gap with optimal parameters.** The colour scale shows the ratio of the accuracy on hard instances reached by curriculum over no-curriculum (a), anti-curriculum over no-curriculum (b), and curriculum over anti-curriculum (c), as a function of the total dataset size ($\alpha_1 + \alpha_2$) and easy dataset size ($\alpha_1$). Curriculum broadly benefits performance and anti-curriculum is effective in certain regions, but the size of the improvement is modest. Parameters: $\rho = 0.50, \Delta_1 = 0, \Delta_2 = 1$.

learning is best. Interestingly, there is a sizeable region of the diagram in which *both* curriculum and anti-curriculum help, possibly explaining why both have been recommended in prior work [12, 14, 43, 44, 45]. A possible intuition behind this counter-intuitive phenomenon highlighted by our analysis is that, in some settings, the large amount of noise contained in the hard data will always be too disruptive for effective learning. Thus, leaving the easy (cleaner) data for last could allow the model to better exploit it.

Further, we find that our setting, in which a small task-relevant signal is embedded in large task-irrelevant variation, is critical to the benefit of curriculum. Fig. 4 shows performance as a function of sparsity $\rho$, additional details are deferred in the SM **??**. Non-sparse tasks do not benefit. Hence curriculum aids tasks with many irrelevant factors of variation. Interestingly, the literature from human psychology shows precisely this: no curriculum benefits for low-dimensional tasks or tasks with no variation in irrelevant dimensions [6].

Our results also highlight the intricate dependence of curriculum on parameters of the learning setup. If not all parameters are correctly optimised, we can observe more complex scenarios. For instance, the initialisation condition for the norm of the weights of the student plays an important role. We explore this dependence by changing the variance of the normal distribution from which the initial weights are sampled from. We observe that anti-curriculum learning becomes the best strategy when the variance is large, as shown in Fig. 3 for weights of order 1. In this case, curriculum learning shows an advantage only in the first phase when easy examples are shown, which is consistent with the results of [19]. However, in the next phase when hard examples are shown, the curriculum strategy does not extract enough information and it is outperformed by the other two strategies. The fact that curriculum or anti-curriculum can look better depending on the parameter setting might help explain the confusion in the literature over the best protocol [12, 14, 43, 44, 45]. At least in this model, better performance from anti-curriculum is a signature of a sub-optimal choice of the parameters.

To summarise our findings in this online learning setting, curriculum mainly offers a *dynamical advantage*: it speeds up learning but has minimal impact on asymptotic performance.

## 4 Batch learning solution

The previous section discussed the online case where each example is used once and then discarded. However, in common machine learning practice, neural networks typically revisit each sample repeatedly until convergence. Therefore an important question is: *can curricula lead to a generalisation improvement when trained on the same dataset until convergence?*

We investigate this question by considering a student that learns from slices of a dataset in distinct optimisation phases, where in each phase the student optimises a $L_2$-regularised logistic loss. Without

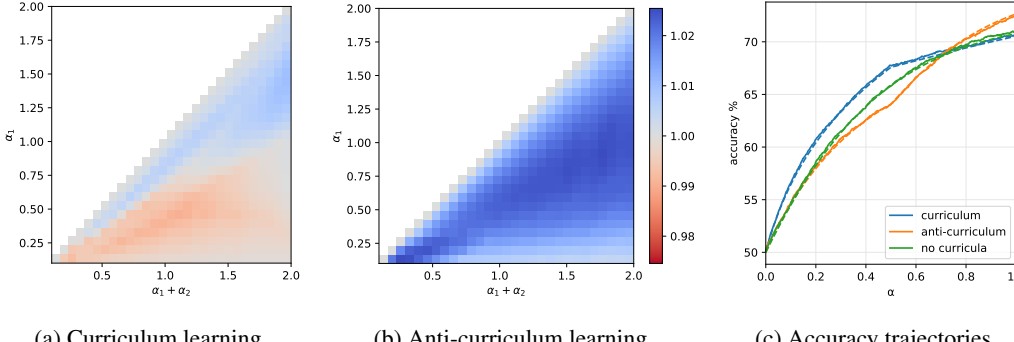

(a) Curriculum learning.  (b) Anti-curriculum learning.  (c) Accuracy trajectories.

Figure 3: **Performance gap starting from high initialisation norm.** The first two figures show the accuracy-gap on hard instances between curriculum learning and the baseline (a) and anti-curriculum learning and the baseline (b). Contrary to the phase diagram in Fig. 2, curriculum learning is not always the optimal and anti-curriculum is not always the worst strategy. The right panel shows the accuracy evaluated on the hard samples for $\alpha_1 = \alpha_2 = 0.5$.

further modification, curriculum can have no effect in this setting: due to the convex nature of the teacher-student setup [22], the network is bound to converge to a minimum uniquely determined by the final slice of data, with no memory of the progress made at intermediate steps. This simple observation may help explain empirical observations on real data, such as [15], which find no benefit of curriculum in standard settings. In fact, in principle curriculum could still influence non-convex problems [12] but empirical results in the ML field are not showing clear signals of memory retention. A possible explanation of this is that relying on dynamical memory effects requires careful tuning of the learning rate and of the number of training epochs, while typical choices for these hyper-parameters could lead to memory loss and performance inconsistencies. These observations raise the theoretical question of how to better implement curriculum learning to induce a non-vanishing effect also in batch learning settings.

To instantiate a long-term memory effect in our model, we propose biasing the optimisation landscape via a Gaussian prior, centred around the optimiser of the previous learning phase. The additional term in the loss acts as an elastic coupling between the successive phases, and the associated intensity $\gamma_{12}$ is then an additional hyper-parameter of the model. This scheme is similar to regularisation methods proposed against catastrophic interference in continual learning, such as Synaptic Intelligence [46]. Changing the loss according to the curriculum prescription effectively makes the learning algorithm *aware* of the different levels of difficulty in the dataset.

Tools from statistical physics can be used to analytically compute test performance under this scheme. In order to simplify the presentation, we first consider just two learning phases. It is natural to frame this setting as a 2-level problem, involving two systems with independent copies of the network weights $\boldsymbol{W}_1$ and $\boldsymbol{W}_2$. In a typical statistical physics approach, we associate a Boltzmann-Gibbs measure to the systems, with an energy function determined by the regularised logistic loss $\mathcal{L}_\gamma$. While the statistical properties of the first system can be determined self-consistently, the added elastic interaction creates a dependence of the second measure on the configurations of the first system. In mathematical terms, the coupled system is represented by the following partition function:

$$\langle Z(\boldsymbol{W}_2, \boldsymbol{W}_1; \mathcal{D}_1, \mathcal{D}_2)\rangle_{\boldsymbol{W}_1} = \int d\boldsymbol{W}_1 \frac{e^{-\beta_1 \mathcal{L}_{\gamma_1}(\boldsymbol{W}_1, \mathcal{D}_1)}}{Z_1(\boldsymbol{W}_1)} \log \int d\boldsymbol{W}_2 \, e^{-\beta_2 \left( \mathcal{L}_{\gamma_2}(\boldsymbol{W}_2, \mathcal{D}_2) + \frac{\gamma_{12}}{2}\|\boldsymbol{W}_2 - \boldsymbol{W}_1\|_2^2 \right)}$$

(3)

where $\mathcal{D}_1, \mathcal{D}_2$ denote the two dataset slices. This object represents the normalisation of the Boltzmann-Gibbs measure, and allows one to extract relevant information on the asymptotic behaviour of our model. The optimisations entailed in each learning phase can be described in the "low noise" limit of $\beta_1, \beta_2 \to \infty$, where the measures focus on the minimisers of the respective losses. In order to study a self-averaging quantity that does not depend on a specific realisation of the dataset, we aim to compute the associated average free-energy:

$$\Phi = \lim_{N \to \infty} \lim_{\beta_1, \beta_2 \to \infty} \frac{1}{\beta_2 N} \left\langle \log \langle Z(\boldsymbol{W}_2, \boldsymbol{W}_1; \mathcal{D}_1, \mathcal{D}_2)\rangle_{\boldsymbol{W}_1} \right\rangle_{\mathcal{D}_1, \mathcal{D}_2}.$$

(4)

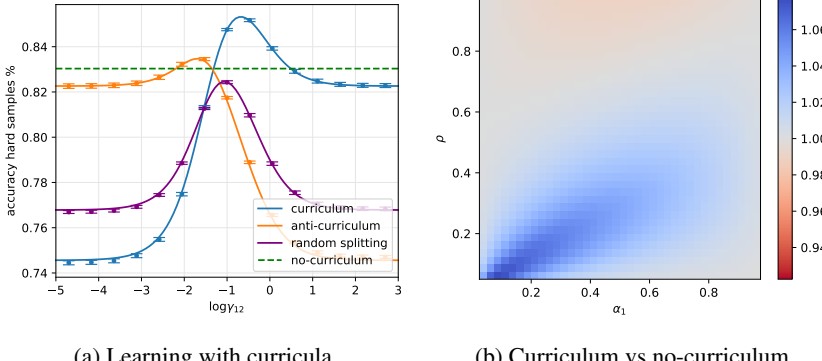

(a) Learning with curricula

(b) Curriculum vs no-curriculum

Figure 4: **Effect of elastic coupling (Gaussian prior) between curriculum phases.** (a) comparison between asymptotic performance of curricula (full lines) and single batch learning, at $\alpha_1 = 1$ $\alpha_2 = 1$, with a regularisation $\gamma_1$ that yields the best generalisation when learning the entire dataset (in principle not optimal for the other strategies). The points represent the results from 10 numerical simulations at size $N = 2000$. Parameters: $\rho = 0.50$, $\Delta_1 = 0$ and $\Delta_2 = 1$. (b) ratio between the accuracy reached by curriculum learning over anti-curriculum as a function of the number of easy samples in a dataset of dimension $\alpha_1 + \alpha_2 = 1$, and of the sparsity level of the teacher $\rho$. Note that $\rho$ can also be seen as the fraction of relevant components in the inputs. $\Delta_1 = 0$ and $\Delta_1 = 1$. $\gamma_1 = \gamma_2$ and $\gamma_{12}$ where set the values that optimise test performance.

This quantity can be seen as a special case of the so-called Franz-Parisi potential computation [47, 48], and the entailed double average can be evaluated through the replica method. Refer to SM for details.

Similar to the online case, in high-dimensions the free-entropy concentrates on a deterministic function that depends on several order parameters that capture the geometrical distribution of teacher and student configurations. In addition to those already introduced in Sec. 3, we also have $\delta Q$, which is linked to the variance of the student norm. Moreover, for each order parameter we also need to introduce a conjugate parameter, denoted in the following with the hat symbol. The final expression for the free-energy reads:

$$\Phi = \text{extr}\Big[ - \Big( \hat{R}R + \frac{1}{2} \Big( \big( \hat{Q}\delta Q - \delta\hat{Q}Q \big)_{r+i} \Big) \Big) + g_S(\gamma_1, \gamma_2, \gamma_{12}) + \alpha_1 \, g_E \, (\Delta_1) + \alpha_2 \, g_E \, (\Delta_2) \Big]$$
(5)

where $g_S$ and $g_E$ are two scalar functions, often called entropic and energetic channels, that encode the dependence of the optimisation problem on the Gaussian prior and the logistic loss respectively. The extremum condition for the free-energy yields a system of fixed-point equations that converge to an asymptotic prediction for the order parameters, comparable with the results of numerical simulations on large instances, Fig. 4. At convergence, the order parameters can be inserted again in Eq. 2 to obtain an estimate of the test accuracy. Note that this formalism is not limited to two phases, but can be extended to the case of a discrete number of sequential stages.

**The importance of sparsity.** Sparsity is a key ingredient in determining the impact of curriculum strategies. It naturally introduces a notion of relevant and irrelevant inputs, and defines a secondary learning goal: identifying what part of the presented data should be disregarded by the model. Curriculum learning can aid this identification process, since the easy samples are more transparent to this structure. This is also observed in human experiments [6]. However, the relative difficulty of the problem of inferring the support of the teacher and the problem of aligning with its non-zero components depends on the degree of sparsity $\rho$, so the effectiveness of curriculum can vary with it.

In the right panel of Fig. 4, we explore the interplay between the sparsity of the teacher $\rho$ and the fraction of easy samples in the dataset $\alpha_1$, comparing curriculum with the no-curriculum baseline. The phase diagram highlights the variability in the impact of the curriculum ordering:

- Curriculum is most effective at low values of $\rho$ and close to the diagonal, where the fraction of easy examples in the dataset is comparable to the fraction of relevant dimensions.

- When $\rho > 0.5$, the possible gain from ordering the samples according to difficulty is counterbalanced by the instrinsic cost of splitting the information content into two blocks, thus curriculum can become detrimental.

- When $\alpha_1$ is too small compared to $\rho$ (above diagonal), the first stage in the curriculum strategy can only help in the support identification problem, but will not allow a good estimation of the direction of the teacher. Because of the elastic prior, the second stage cannot improve too much over it and the effect of curriculum is small.

- When $\alpha$ is larger than the sparsity (below diagonal), the easy examples contain sufficient information for solving both the support and the teacher estimation problems, and this information is also exploited by the baseline. Thus the improvement of curriculum becomes negligible.

We refer to the SM for an in-depth comparison with anti-curriculum.

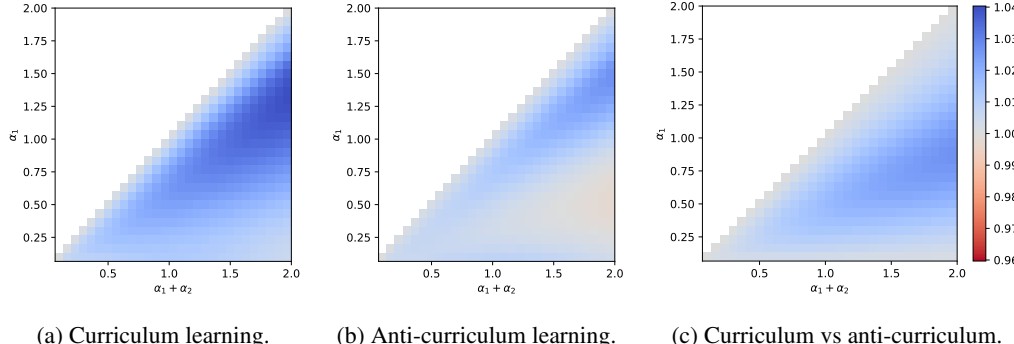

(a) Curriculum learning.      (b) Anti-curriculum learning.      (c) Curriculum vs anti-curriculum.

Figure 5: **Phase diagram for the performance gap in the batch setting.** The colour scale shows the ratio of the accuracy on hard instances for curriculum over no-curriculum (a), anti-curriculum over no-curriculum (b), and curriculum over anti-curriculum (c), as a function of the total dataset size ($\alpha_1 + \alpha_2$) and easy dataset size ($\alpha_1$). In contrast to the online case, performance benefits are greater and curriculum is strictly better than anti-curriculum. Both $\gamma_1 = \gamma_2$ and $\gamma_{12}$ are optimised point-wise, in order to yield the best test accuracy. Parameters: $\rho = 0.50, \Delta_1 = 0, \Delta_2 = 1$.

**Asymptotic advantages of curriculum.**    Contrary to the case of online SGD, if the fraction of relevant directions is small, batch learning with elastic coupling notably improves test accuracy of both curriculum and anti-curriculum above the baseline. This confirms the utility of curriculum strategies when the signal is partially "hidden in clutter" [49].

Fig. 5 shows similar phase diagrams to Fig. 2 but for the batch setting. At each point in the phase diagram the regularisation level $\gamma_1 = \gamma_2$ and the coupling $\gamma_{12}$ are optimised to yield the best accuracy. We find that the performance order is nearly always preserved: curriculum followed by anti-curriculum followed by baseline. In the SM we see similar improvements by applying the elastic coupling strategy both in the online setting and on real data.

In summary, in the batch setting, splitting the learning process in stages might not be advantageous per se. However, our observations show that if the loss is modified to reduce memory loss between the learning stages, curriculum learning strategies can offer a measurable *asymptotic advantage*.

## 5   Connection with experimental literature

Recent work has suggested that curriculum learning could provide an important window into the learning algorithms at work in biology [51]. Our analysis makes several predictions for curriculum effects. In this section we assess these predictions based on connections to extant experiments and propose future experimental tests.

First, we find that a curriculum strategy yields a speed up in learning in all the tested settings (see Fig. 1c). This acceleration is broadly consistent with the findings from cognitive science [1, 2, 6]. By contrast, our results show that the speed improvement does not necessarily translate into a sizeable

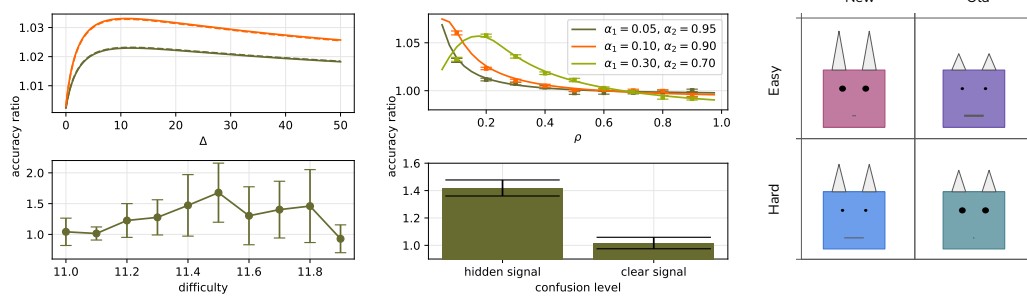

(a) Generalisation gap per difficulty.    (b) Role of sparsity.    (c) Fading experiment [6].

Figure 6: **Connection with psychology experiments.** (a) Top: Accuracy ratio of different strategies in the model, with curriculum/no-curriculum in green and curriculum/anti-curriculum in orange. The ratio shows non-monotonic behaviour. Bottom: The accuracy ratio obtained by [50]. Parameters $\rho = 0.5$, $\Delta_1 = 0.0$, $\Delta_2 = 1.0$, $\alpha_1 = 1$, $\alpha_2 = 1$ and optimal learning rate, norm at initialisation and weight decay intensity. (b) Top: Dependence on the sparsity of the generalisation gain of curriculum over no-curriculum, measured as ratio between final accuracy, for fixed total dataset size ($\alpha_1 + \alpha_2 = 1$). Bottom: The ratio obtained from experiments 3 and 4 of [6]. (c) Example cartoon stimuli from the "fading" paradigm used in [6], where participants distinguish daemons of the old world from daemons of the new world. The distinguishing feature (horn length) is diluted among many irrelevant features (colour, eye size, mouth size). Highlighting the relevant feature to participants leads to better and faster learning.

generalisation error improvement, and the performance achieved at the end of training can even deteriorate when learning hyperparameters are not fully optimised (c.f. Fig. 3). Deterioration due to curricula has generally not been reported in the psychology literature, though it has been observed in ML [15]. This fact may suggest that animals naturally learn with near-optimal hyperparameters such that curricula generally confer benefits.

A more specific observation concerns the performance on different difficulties after learning. As reported in [50], human and rodent subjects trained in an auditory task using curricula showed the greatest improvement for intermediate levels of difficulty as depicted in Fig. 6a bottom panel. The same conclusion can be drawn from the experiment of [7, 8], where, surprisingly, subjects trained with curricula to classify medical images showed poor performance in hard tasks compared to the control group. To address this phenomenon, we calculate accuracy as a function of difficulty in the model in Fig. 6a top panel. Consistent with these experiments, we find regimes where the gap between curriculum learning and the baseline is non-monotonic, with the largest performance gain for intermediate difficulties. Contrary to [7, 8], however, we do not observe negative effects of curriculum for high difficulties. Further experiments that more systematically manipulate training and transfer difficulties could provide a stronger test of these predictions.

A key ingredient in our model is the role of sparsity, such that a small signal is embedded amidst many irrelevant features. Experimentally, the importance of having many factors of variation to obtaining a curriculum effect has been documented in the "fading" experiments of [6]. Human subjects were trained on classification tasks involving stimuli with one task-relevant feature dimension and a variable number of task-irrelevant feature dimensions. Example cartoon "daemon" stimuli are depicted in Fig. 6c, where for instance horn height might be the distinguishing feature while colour, eye size, and mouth size might constitute task-irrelevant features. Without any irrelevant factors of variation ($\rho = 1$), they report no curriculum benefit. By contrast when $75\%$ of features are irrelevant ($\rho = .25$), they record a strong curriculum effect, as shown in Fig. 6b bottom. This qualitative trend is also observed in our model (Fig. 6b top). While these experiments tested only two sparsity levels, further experiments could sample this dimension more extensively and test for interactions with the fraction of easy and hard examples. We note that while the connectionist literature has addressed the effect of curriculum in several settings [39, 40, 10, 11], we found that easy-to-hard effects appear even in a simple setup without need for complex networks and/or dynamics.

Finally, our results may shed light on self-generated curricula during human development [52, 53]. Children undergo a vocabulary spurt that coincides with their ability to grasp and centre objects in the

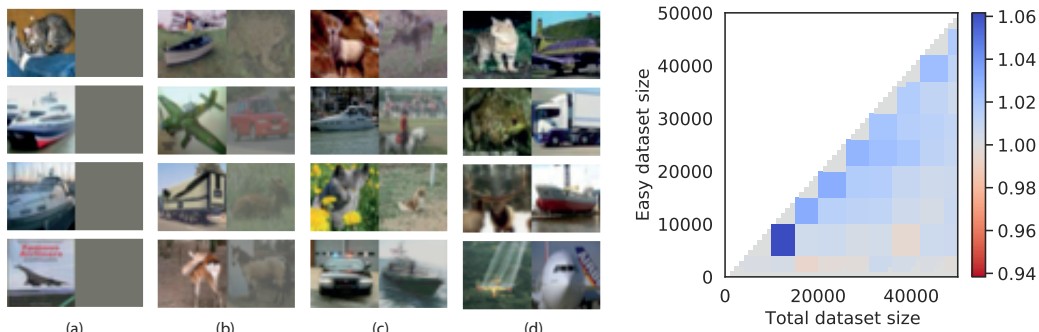

Figure 7: **Experimental setting on CIFAR10-derived data.** (a) Input samples combine a task-relevant image with a distractor image, and become progressively harder from left to right. (b) Ratio between final accuracy on hard instances for curriculum learning versus no curriculum. $\eta, \gamma, \gamma_{12}$, init, and stopping time are optimised.

visual field [53]. Quantitative estimates of the amount of clutter (irrelevant objects) in self-generated views decrease due to this grasping ability, yielding a self-generated curriculum [49, 54]. Our model similarly predicts that reducing clutter should improve learning speed and performance.

**Real-World Demonstration.** To verify this prediction in a richer visual setting, we construct a simple cluttered object classification task from the CIFAR10 dataset [55] by patching two images together into a $32 \times 64$ input image (Fig. 7a). The task is to produce the class label of the image on the left. The right image is a distractor that is irrelevant to the classification. To vary difficulty, we scale the contrast of the irrelevant image (Fig. 7a-d). We train a single-layer network with the cross-entropy loss and the curriculum protocol with Gaussian prior between two curriculum stages, implemented in Pytorch Lightning to ensure that training parameters accord with standard practice. We optimised hyperparameters in each curriculum phase separately. We trained all combinations of five elastic penalties log spaced between $1e-3$ and $1e2$, and weight decay parameters $\{0, .2, .5\}$. We then compute the best performing model for five random seeds and take the mean over seeds. Further dataset, model and experimental details are given in Appendix **??**. As shown in Fig. 7b, curriculum improves performance, particularly when easy examples make up a large proportion of the dataset, confirming that curricula that reduce clutter can benefit learning.

## 6 Conclusions

We analysed a model of curriculum learning introduced by [12] and amenable of analytical treatment. This simple setting sheds light on results observed in the cognitive science and machine learning literature, and the theoretical tractability allows for exploration of a wide range of parameters that would be costly to obtain through experiments. Future work will need to move beyond models with simple loss landscapes to address the impact of curricula in complex tasks like reinforcement learning. Nevertheless, the model recapitulates a variety of observations in the literature [50, 56, 57], revealing that easy-to-hard effects can appear when a sparse signal is embedded in many irrelevant dimensions of variation. We find that making the algorithm curriculum-aware by modifying the loss can better exploit curricula, offering a potential route for improved practical algorithms. Other curriculum-aware approaches are possible such as adapting the learning algorithm [58] or the architecture [10]. On the psychology side, our predictions can help in designing new experiments, for instance testing the counter-intuitive benefit of anti-curriculum learning for intermediate sparsity.

## Acknowledgments and Disclosure of Funding

We thank Miguel Ruiz-Garcia and Ronald Dekker for important discussions. L.S. acknowledges funding from the ERC European Union Horizon 2020 Research and Innovation Program Grant Agreement 714608-SIiLe. S.S.M. & A.S. were supported by a Wellcome and Royal Society Henry Dale Fellowship (216386/Z/19/Z) and Sainsbury Wellcome Centre Core Grant (219627/Z/19/Z, GAT3755). A.S. is a CIFAR Azrieli Global Scholar in the Learning in Machines & Brains programme.

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
