# Supplemental Material

## A  State evolution of the online dynamics

In this section we show how to derive the dynamical equations for the online dynamics. The equations given in an implicit form in the main text, $f_{Q_r}, f_{Q_i}, f_R$, are reported explicitly at the end of the next section, Eqs. (A.23-A.25). Finally, in the subsequent section, we comment on how the state evolution is modified to deal with the Gaussian priors and we derive the new dynamical equations for that case.

**Derivation**  We follow the derivation proposed in [26, 42] to derive the averaged high-dimensional dynamical equations. The student is a 1-layer network that minimises sample-wise the square error

$$\mathcal{L}^\mu = \frac{1}{2}\left(y^\mu - \hat{y}^\mu\right)^2 \doteq \frac{1}{2}\left(\delta^\mu\right)^2. \tag{A.1}$$

Given $\phi(\cdot) = \mathrm{sign}(\cdot), \sigma(\cdot) = \mathrm{erf}(\cdot/\sqrt{2})$, the online stochastic gradient descent updates are

$$\boldsymbol{W}^{\mu+1} = \boldsymbol{W}^\mu - \frac{\eta}{\sqrt{N}}\sigma'(\lambda_r^\mu + \lambda_i^\mu)\delta^\mu \boldsymbol{x}^\mu, \tag{A.2}$$

with

$$\lambda_r^\mu = \frac{1}{\sqrt{N}}\boldsymbol{W}_r \cdot \boldsymbol{x}_r^\mu, \tag{A.3}$$

$$\lambda_i^\mu = \frac{1}{\sqrt{N}}\boldsymbol{W}_i \cdot \boldsymbol{x}_i^\mu, \tag{A.4}$$

$$\rho^\mu = \frac{1}{\sqrt{N}}\boldsymbol{W}_T \cdot \boldsymbol{x}_r^\mu. \tag{A.5}$$

The evolution of the dynamics can be tracked using 4 order parameters:

$$Q_r = \frac{1}{N}\boldsymbol{W}_r \cdot \boldsymbol{W}_r, \tag{A.6}$$

$$Q_i = \frac{1}{N}\boldsymbol{W}_i \cdot \boldsymbol{W}_i, \tag{A.7}$$

$$R = \frac{1}{N}\boldsymbol{W}_r \cdot \boldsymbol{W}_T, \tag{A.8}$$

$$T = \frac{1}{N}\boldsymbol{W}_T \cdot \boldsymbol{W}_T; \tag{A.9}$$

representing the overlaps between the weights of student (relevant and irrelevant parts) and teacher.

The evolution of those follow from the definition of the dynamics Eq. (A.2). In the high-dimensional limit the random variables in the problem concentrates around the mean, therefor to the leading order we have the following equations

$$Q_r[k+1] = Q_r[k] + \frac{1}{N}\left[2\eta\mathbb{E}[\delta\,\sigma'(\lambda_r + \lambda_i)\lambda_r] + \rho\Delta\eta^2\mathbb{E}[\delta^2\,\sigma'(\lambda_r + \lambda_i)^2]\right]; \tag{A.10}$$

$$Q_i[k+1] = Q_r[k] + \frac{1}{N}\left[2\eta\mathbb{E}[\delta\,\sigma'(\lambda_r + \lambda_i)\lambda_i] + (1-\rho)\Delta\eta^2\mathbb{E}[\delta^2\,\sigma'(\lambda_r + \lambda_i)^2]\right]; \tag{A.11}$$

$$T[k+1] = Q_r[k] + \frac{1}{N}\left[\eta\mathbb{E}[\delta\,\sigma'(\lambda_r + \lambda_i)\rho]\right]. \tag{A.12}$$

Where the expectation acts with respect to all the stochastic variables. In order to obtain explicit formulae we need to evaluate those averages. The random variables in the equations $-\lambda_r$, $\lambda_i$ and $\rho-$ are Gaussian with zero mean, to characterise them we only need their covariance:

$$\Sigma_{\lambda_r,\lambda_i,\rho} = \begin{pmatrix} Q_r & 0 & R \\ 0 & Q_i & 0 \\ R & 0 & T \end{pmatrix}.$$

In order to derive analytical expression we must evaluate the expected values: $\mathbb{E}[\phi(\rho)\sigma'(\lambda)\rho]$, $\mathbb{E}[\phi(\rho)\sigma'(\lambda)\lambda]$, $\mathbb{E}[\sigma(\lambda)\sigma'(\lambda)\rho]$, $\mathbb{E}[\sigma(\lambda)\sigma'(\lambda)\lambda]$, $\mathbb{E}[\phi(\rho)^2\sigma'(\lambda)^2]$, $\mathbb{E}[\sigma(\lambda)^2\sigma'(\lambda)^2]$, and $\mathbb{E}[\phi(\rho)\sigma(\lambda)\sigma'(\lambda)^2]$. Where $\sigma$ is the activation function of the student and $\phi$ is the activation function of the teacher (in particular $\phi(\cdot) = \mathrm{sign}(\cdot)$ for classification).

$$\mathbb{E}[\phi(\rho)\sigma'(\lambda)\rho] = \frac{2}{\pi} \frac{\sqrt{T(Q_r + Q_i + 1) - R^2}}{Q_r + Q_i + 1} \tag{A.14}$$

$$\mathbb{E}[\phi(\rho)\sigma'(\lambda)\lambda_r] = \frac{2}{\pi} \frac{R(Q_i + 1)}{Q_r + Q_i + 1} \frac{1}{\sqrt{T(Q_r + Q_i + 1) + R^2}}. \tag{A.15}$$

$$\mathbb{E}[\phi(\rho)\sigma'(\lambda)\lambda_i] = -\frac{2}{\pi} \frac{RQ_i}{Q_r + Q_i + 1} \frac{1}{\sqrt{T(Q_r + Q_i + 1) + R^2}}. \tag{A.16}$$

$$\mathbb{E}[\sigma(\lambda)\sigma'(\lambda)\rho] = \frac{2}{\pi} \frac{R}{Q_r + Q_i + 1} \sqrt{\frac{Q_i + 1}{2Q_i^2 + 2Q_rQ_i + 3Q_i + 2Q_r + 1}}. \tag{A.17}$$

$$\mathbb{E}[\sigma(\lambda)\sigma'(\lambda)\lambda_r] = \frac{2}{\pi} \frac{Q_r}{Q_r + Q_i + 1} \sqrt{\frac{Q_i + 1}{2Q_i^2 + 2Q_rQ_i + 3Q_i + 2Q_r + 1}}. \tag{A.18}$$

$$\mathbb{E}[\sigma(\lambda)\sigma'(\lambda)\lambda_i] = \frac{2}{\pi} \frac{Q_i}{Q_r + Q_i + 1} \sqrt{\frac{Q_r + 1}{2Q_r^2 + 2Q_rQ_i + 3Q_r + 2Q_i + 1}}. \tag{A.19}$$

$$\mathbb{E}[\phi(\rho)^2\sigma'(\lambda)^2] = \frac{2}{\pi} \frac{1}{\sqrt{2Q_r + 2Q_i + 1}}. \tag{A.20}$$

$$\mathbb{E}[\sigma(\lambda)^2\sigma'(\lambda)^2] = \frac{4}{\pi^2} \frac{1}{\sqrt{1 + 2(Q_r + Q_i)}} \sin^{-1}\left(\frac{Q_r + Q_i}{1 + 3(Q_r + Q_i)}\right). \tag{A.21}$$

$$\mathbb{E}[\phi(\rho)\sigma(\lambda)\sigma'(\lambda)^2] = \frac{4}{\pi^2} \frac{1}{\sqrt{2(Q_r + Q_i) + 1}}$$
$$\sin^{-1}\left(\frac{R\sqrt{Q_r + Q_i}}{\sqrt{3(Q_r + Q_i) + 1}\sqrt{(2Q_r + 2Q_i + 1)[T(Q_r + Q_i) - R^2] + R^2}}\right). \tag{A.22}$$

Finally, we can substitute those equations into the Eqs. (A.10-A.12) and obtained the state evolution equations used in the main Sec. 3:

$$f_{Q_r}(Q_r[k], Q_i[k], R[k], T) = (1 - \eta\gamma)^2 Q_r[k] + \frac{4\eta(1 - \eta\gamma)}{N\pi(Q_r[k] + \Delta Q_i[k] + 1)} \times$$
$$\left[\frac{R[k](\Delta Q_i[k] + 1)}{\sqrt{T(Q_r[k] + \Delta Q_i[k] + 1) + R[k]^2}} - \frac{Q_r[k]}{\sqrt{2Q_r[k] + 2\Delta Q_i[k] + 1}}\right]$$
$$+ \frac{4}{\pi^2} \frac{\rho\eta^2}{N\sqrt{2(Q_r[k] + \Delta Q_i[k]) + 1}} \left[\frac{\pi}{2} + \sin^{-1}\left(\frac{Q_r[k] + \Delta Q_i[k]}{1 + 3(Q_r[k] + \Delta Q_i[k])}\right) + \right.$$
$$\left. - 2\sin^{-1}\left(\frac{R[k]}{\sqrt{3(Q_r[k] + \Delta Q_i[k]) + 1}\sqrt{T(2Q_r[k] + 2\Delta Q_i[k] + 1) - 2R[k]^2}}\right)\right]; \tag{A.23}$$

$$f_{Q_i}\big(Q_r[k], Q_i[k], R[k], T\big) = (1 - \eta\gamma)^2 Q_i[k] - \frac{4\eta(1 - \eta\gamma)\Delta Q_i[k]}{N\pi(Q_r[k] + \Delta Q_i[k] + 1)} \times$$

$$\left[ \frac{R[k]}{\sqrt{T(Q_r[k] + \Delta Q_i[k] + 1) + R[k]^2}} + \frac{1}{\sqrt{2Q_r[k] + 2\Delta Q_i[k] + 1}} \right] +$$

$$+ \frac{4}{\pi^2} \frac{(1 - \rho)\Delta\eta^2}{N\sqrt{2(Q_r[k] + \Delta Q_i[k]) + 1}} \left[ \frac{\pi}{2} + \sin^{-1}\left( \frac{Q_r[k] + \Delta Q_i[k]}{1 + 3(Q_r[k] + \Delta Q_i[k])} \right) + \right.$$

$$\left. - 2\sin^{-1}\left( \frac{R[k]}{\sqrt{3(Q_r[k] + \Delta Q_i[k]) + 1}\sqrt{T(2Q_r[k] + 2\Delta Q_i[k] + 1) - 2R[k]^2}} \right) \right]; \tag{A.24}$$

$$f_R\big(Q_r[k], Q_i[k], R[k], T\big) = (1 - \eta\gamma)R[k] + \frac{2\eta}{N\pi(Q_r[k] + \Delta Q_i[k] + 1)} \times$$

$$\left[ \frac{T(Q_r[k] + \Delta Q_i[k] + 1) - R[k]^2}{\sqrt{T(Q_r[k] + \Delta Q_i[k] + 1) - R[k]^2}} - \frac{R[k]}{\sqrt{2Q_r[k] + 2\Delta Q_i[k] + 1}} \right]. \tag{A.25}$$

**Elastic coupling** The introduction of the elastic coupling between stages of learning adds five new order parameters: three of them are just reminder of the previous stage and do not need to by updated $\tilde{Q}_r = \boldsymbol{W}_1^r \cdot \boldsymbol{W}_1^r / N$, $\tilde{Q}_i = \boldsymbol{W}_1^i \cdot \boldsymbol{W}_1^i / N$, and $\tilde{R} = \boldsymbol{W}_1^i \cdot \boldsymbol{W}^T / N$; two measure the correlation between the two stages $S_r = \boldsymbol{W}_1^r \cdot \boldsymbol{W}_2^r / N$ and $S_i = \boldsymbol{W}_1^i \cdot \boldsymbol{W}_2^i / N$ to the equations. These terms have associated their own state evolution equations slightly modified the updates of the other order parameters.

$$Q_r[k+1] = (1 - \eta\gamma + \eta\gamma_{12})^2 Q_r[k] + \frac{2\eta}{N}(1 - \eta\gamma + \eta\gamma_{12})\mathbb{E}[\delta\,\sigma'(\lambda_r + \lambda_i)\lambda_r]$$

$$+ \rho\Delta\frac{\eta^2}{N}\mathbb{E}[\delta^2\,\sigma'(\lambda_r + \lambda_i)^2] + 2\eta\gamma_{12}(1 - \eta\gamma + \eta\gamma_{12})S_r[k] + \eta^2\gamma_{12}^2\tilde{Q}_r[k] \tag{A.26}$$

$$- \frac{2\eta^2\gamma_{12}}{N}\mathbb{E}[\delta\,\sigma'(\lambda_r + \lambda_i)\tilde{\lambda}_r];$$

$$Q_i[k+1] = (1 - \eta\gamma + \eta\gamma_{12})^2 Q_i[k] + \frac{2\eta}{N}(1 - \eta\gamma + \eta\gamma_{12})\mathbb{E}[\delta\,\sigma'(\lambda_r + \lambda_i)\lambda_i]$$

$$+ (1 - \rho)\Delta\frac{\eta^2}{N}\mathbb{E}[\delta^2\,\sigma'(\lambda_r + \lambda_i)^2] + 2\eta\gamma_{12}(1 - \eta\gamma + \eta\gamma_{12})S_i[k] \tag{A.27}$$

$$+ \eta^2\gamma_{12}^2\tilde{Q}_i[k] - \frac{2\eta^2\gamma_{12}}{N}\mathbb{E}[\delta\,\sigma'(\lambda_r + \lambda_i)\tilde{\lambda}_i];$$

$$R[k+1] = (1 - \eta\gamma + \eta\gamma_{12})R[k] + \frac{\eta}{N}\mathbb{E}[\delta\,\sigma'(\lambda_r + \lambda_i)\rho] - \eta\gamma_{12}\tilde{R}[k]; \tag{A.28}$$

$$S_r[k+1] = (1 - \eta\gamma + \eta\gamma_{12})S_r[k] + \frac{\eta}{N}\mathbb{E}[\delta\,\sigma'(\lambda_r + \lambda_i)\tilde{\lambda}_r] - \eta\gamma_{12}\tilde{Q}_r[k]; \tag{A.29}$$

$$S_i[k+1] = (1 - \eta\gamma + \eta\gamma_{12})S_i[k] + \frac{\eta}{N}\mathbb{E}[\delta\,\sigma'(\lambda_r + \lambda_i)\tilde{\lambda}_i] - \eta\gamma_{12}\tilde{Q}_i[k]. \tag{A.30}$$

Introduced $\tilde{\lambda}_r = \frac{1}{\sqrt{N}}\boldsymbol{x}_r \cdot \tilde{\boldsymbol{W}}_r$ and $\tilde{\lambda}_i = \frac{1}{\sqrt{N}}\boldsymbol{x}_i \cdot \tilde{\boldsymbol{W}}_i$, this two additional random variables need to be averaged together with the others. The joint distribution of $\lambda_r, \lambda_i, \tilde{\lambda}_r, \tilde{\lambda}_i, \rho$ is still Gaussian with zero mean and covariance

$$\Sigma_{\lambda_r, \lambda_i, \tilde{\lambda}_r, \tilde{\lambda}_i, \rho} = \begin{pmatrix} Q_r & 0 & \tilde{S}_r & 0 & R \\ 0 & Q_i & 0 & \tilde{S}_i & 0 \\ \tilde{S}_r & 0 & \tilde{Q}_r & 0 & \tilde{R} \\ 0 & \tilde{S}_i & 0 & \tilde{Q}_i & 0 \\ R & 0 & \tilde{R} & 0 & T \end{pmatrix}.$$

Notice that, a part from a slight change of the existing equations, the coupling introduces only two additional integrals $\mathbb{E}[\delta\,\sigma'(\lambda_r + \lambda_i)\tilde{\lambda}_r]$ and $\mathbb{E}[\delta\,\sigma'(\lambda_r + \lambda_i)\tilde{\lambda}_i]$. After long, but straightforward,

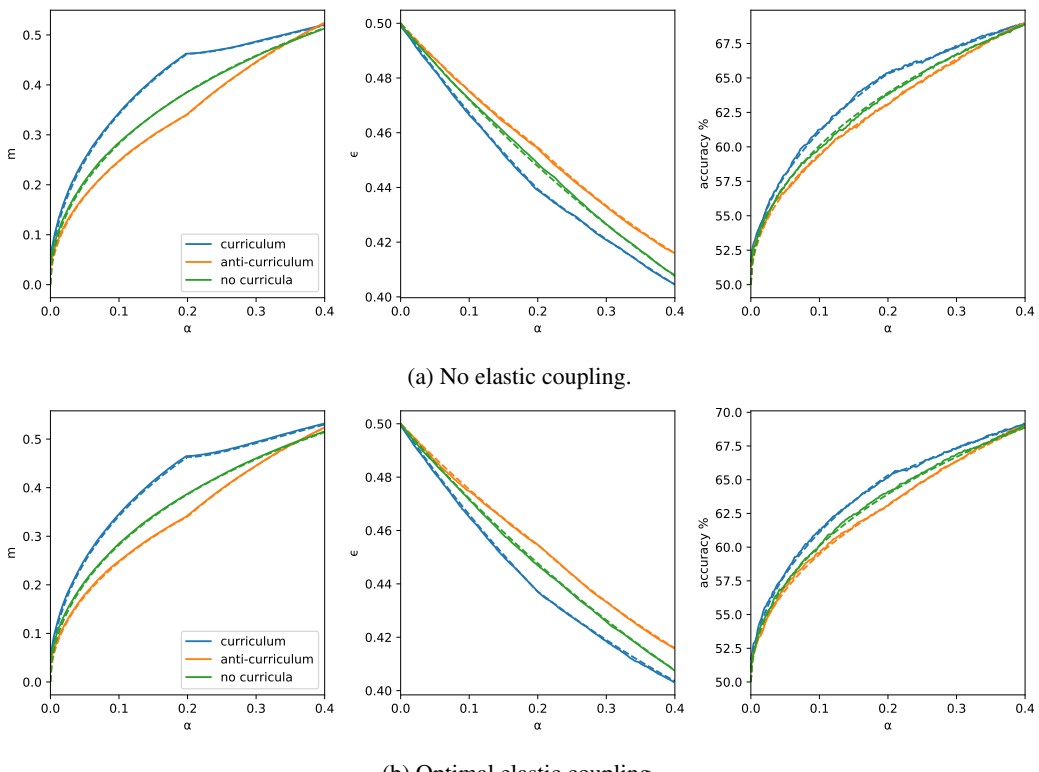

(a) No elastic coupling.

(b) Optimal elastic coupling.

Figure A.1: **Effect of elastic coupling in the curriculum.** Figures showing the teacher-student cosine, the validation loss, and the accuracy of the three learning strategies. The two figures show the performance in presence (above) and absence (below) of elastic coupling. The dashed lines are obtained from the theoretical analysis, the full line come from the average of 500 simulations. The parameters $\eta$, $\gamma$, initialisation are set to the optimal values for each protocol. Parameters: $\rho = 0.5$, $\alpha_1 = 0.2$, $\alpha_2 = 0.2$, $\Delta_1 = 0$, $\Delta_2 = 1$.

computations we obtain

$$
\mathbb{E}[\delta\, \sigma'(\lambda_r + \lambda_i)\tilde{\lambda}_r] = \frac{2}{\pi} \frac{S_r}{Q_r + Q_i + 1} \frac{Q_i + 1}{2Q_i^2 + 2Q_rQ_i + 3Q_i + 2Q_r + 1} +
$$
$$
- \frac{2}{\pi} \frac{TS_r - R\tilde{R}}{Q_rT - R^2} \frac{R(Q_i + 1)}{Q_r + Q_i + 1} \frac{1}{\sqrt{T(Q_r + Q_i + 1) - R^2}} + \tag{A.31}
$$
$$
- \frac{2}{\pi} \frac{T\tilde{R} - RS_r}{Q_rT - R^2} \frac{1}{\sqrt{T(Q_r + Q_i + 1) - R^2}} \frac{1}{\frac{1}{T} + \frac{R^2}{Q_rT - R^2}\left(\frac{1}{T} - \frac{Q_i + 1}{T(Q_r + Q_i + 1) - R^2}\right)},
$$
$$
\mathbb{E}[\delta\, \sigma'(\lambda_r + \lambda_i)\tilde{\lambda}_i] = \frac{2}{\pi} \frac{S_i}{Q_r + Q_i + 1} \frac{Q_r + 1}{2Q_r^2 + 2Q_rQ_i + 3Q_r + 2Q_i + 1} +
$$
$$
- \frac{2}{\pi} \frac{S_i R}{Q_r + Q_i + 1} \frac{1}{\sqrt{T(Q_r + Q_i + 1) - R^2}}. \tag{A.32}
$$

Finally all the expected values are known and we can obtain the analytic updates Eqs. (A.26-A.30) with the coupling. Fig. A.1a shows an instance of the problem at $\alpha_1 = 0.2$ and $\alpha_2 = 0.2$, a situation that is particularly adversarial for curriculum according the phase diagram Fig. 2. This situation is treated by the introduction of Gaussian priors, Fig. A.1b, consistently with the phase diagram in Fig. 7c.

## B Replica computation for the batch case

We here the detailed replica computation employed to obtain the analytic description of curriculum learning in the batch case, in section 4. As mentioned in the main, we aim to study a coupled system, represented by the following partition function:

$$
\langle Z(\boldsymbol{W}_2, \boldsymbol{W}_1; \mathcal{D}_1, \mathcal{D}_2)\rangle_{\boldsymbol{W}_1} = \int d\boldsymbol{W}_1 \frac{e^{-\beta_1 \mathcal{L}_{\gamma_1}(\boldsymbol{W}_1, \mathcal{D}_1)}}{Z_1(\boldsymbol{W}_1)} \log \int d\boldsymbol{W}_2\, e^{-\beta_2\left(\mathcal{L}_{\gamma_2}(\boldsymbol{W}_2, \mathcal{D}_2) + \frac{\gamma_{12}}{2}\|\boldsymbol{W}_2 - \boldsymbol{W}_1\|_2^2\right)},
$$

(B.1)

where the examples is $\mathcal{D}_1, \mathcal{D}_2$ are characterised by a different variances in the irrelevant components.

This type of quantity is usually denoted as a "disordered" partition function in statistical physics jargon, meaning that it is still dependent on a given realisation of the datasets – i.e., the source of disorder in this model. We want to characterise a typical realisation of this object, in the high-dimensional limit. However, because of its long-tailed statistics, the partition function turns out not to be a self-averaging quantity, i.e. its expectation over the dataset realisations will not correspond to the typical case scenario we are after. It is instead better to focus on the computation of the associated average free-entropy:

$$
\Phi = \lim_{N \to \infty} \lim_{\beta_1, \beta_2 \to \infty} \frac{1}{\beta_2 N} \left\langle \log \langle Z(\boldsymbol{W}_2, \boldsymbol{W}_1; \mathcal{D}_1, \mathcal{D}_2)\rangle_{\boldsymbol{W}_1}\right\rangle_{\mathcal{D}_1, \mathcal{D}_2}.
$$

(B.2)

What is immediately apparent is that we have to take the expectation of a logarithm, which is not tractable with rigorous mathematical methods. Moreover, we also have to average over the measure for $\boldsymbol{W}_1$, which is also a complicated operation.

Fortunately, replica theory offers a method for approaching this calculation [47, 48]. The idea is to exploit two separate replica tricks:

- in order to evaluate the disorder average, the logarithm can be removed by replicating the second weight configuration, i.e. introducing $n$ identical replicas $\{\boldsymbol{W}_2^a\}_{a=1}^n$, and extrapolating the final result from the $n \to 0$ limit. This is based on the mathematical identity $\log x = \lim_{n \to 0} \partial_n x^n$.
- the average over the teacher can instead be computed by introducing $\tilde{n} - 1$ non-interacting and a single interacting replica of the first weight configuration $\{\boldsymbol{w}_1^c\}_{c=1}^{\tilde{n}}$. Thus, only the $c = 1$ replica will enter the Gaussian prior in the student measure. The sought statistical average is again recovered in the limit $\tilde{n} \to 0$.

Because of the high-dimensional limit we are considering, all typical realisations of the teacher vector with a given sparsity $\rho$ will yield an identical free-entropy. Thus, we can avoid averaging and instead fix a gauge $\boldsymbol{W}_{T,i} = 1$ for $i = 1, \ldots, \rho N$ and $\boldsymbol{W}_{T,i} = 0$ elsewhere. In order to simplify the presentation, in the following we will assume that the datasets contain respectively $\alpha_1$ and $\alpha_2$ patterns, and that a curriculum ordering was employed, $\Delta_1 < \Delta_2$. Moreover, to avoid confusion with component and replica indices, we will denote with $\tilde{\boldsymbol{W}} = \boldsymbol{W}_1$ and $\boldsymbol{W} = \boldsymbol{W}_2$, so that all quantities with a tilde refer to the optimisation on the first dataset.

After the described replication procedures, we get the following expression for the average free-entropy:

$$
\Phi = \frac{1}{N} \lim_{n, \tilde{n} \to 0} \partial_n \left\langle \lim_{\tilde{\beta}, \beta \to \infty} \frac{1}{\beta} \int \prod_{c=1}^{\tilde{n}} d\tilde{\boldsymbol{W}}^c e^{-\frac{\tilde{\beta}\gamma_1}{2}\|\tilde{\boldsymbol{W}}^c\|_2^2} \prod_{\mu=1}^{\alpha_1 N} \prod_{c=1}^{\tilde{n}} e^{-\frac{\beta}{2}\ell\left(\text{sign}\left(\sum_{i=1}^{\rho N} \frac{x_i^\mu}{\sqrt{N}}\right), \sigma\left(\sum_{i=1}^N \frac{\tilde{W}_i^c x_i^\mu(\Delta_1)}{\sqrt{N}}\right)\right)} \right.
$$

(B.3)

$$
\left. \times \int \prod_{a=1}^n d\boldsymbol{W}^a e^{-\frac{\beta\gamma_2}{2}\|\boldsymbol{W}^a\|_2^2} e^{-\frac{\beta\gamma_{12}}{2}\|\boldsymbol{W}^a - \tilde{\boldsymbol{W}}^1\|_2^2} \prod_{\mu=1}^{\alpha_2} \prod_a e^{-\frac{\beta}{2}\ell\left(\text{sign}\left(\sum_{i=1}^{\rho N} \frac{x_i^\mu}{\sqrt{N}}\right), \sigma\left(\sum_{i=1}^N \frac{W_i^a x_i^\mu(\Delta_2)}{\sqrt{N}}\right)\right)} \right\rangle_{\{\boldsymbol{x}^\mu\}},
$$

where $\ell(y, \hat{y}) = \log(1 + e^{-y\hat{y}})$ indicates the standard logistic loss. The next step is to explicitly compute the averages over the dataset realisations. Before doing that, we need to isolate the dependence of our expression on the patterns, and we achieve this by introducing Dirac's $\delta$-functions for the pre-activations. We will use the integral representation of the $\delta$, with integration variables $u$ for the teacher preactivations $\lambda$ for the student preactivations:

$$
\frac{1}{N} \lim_{n, \tilde{n} \to 0} \partial_n \int \prod_{c=1}^{\tilde{n}} d\tilde{\boldsymbol{W}}^c e^{-\frac{\beta\lambda}{2}\|\tilde{\boldsymbol{W}}^c\|_2^2} \int \prod_{a=1}^{\tilde{n}} d\boldsymbol{W}^a e^{-\frac{\beta\lambda}{2}\|\boldsymbol{W}^a\|_2^2} e^{-\frac{\beta\lambda_{12}}{2}\|\boldsymbol{W}^a - \tilde{\boldsymbol{W}}^1\|_2^2}
$$

(B.4)

$$\times \left\langle \int \prod_\mu \frac{d\tilde{u}_{1\mu} d\hat{\tilde{u}}_{1\mu}}{2\pi} e^{i\hat{\tilde{u}}_{1\mu}\left(\tilde{u}_{1\mu}-\sum_{i=1}^{\rho N}\frac{(\tilde{x}_1)_i^\mu}{\sqrt{N}}\right)} \int \prod_{\mu,c} \frac{d\tilde{\lambda}_{1\mu}^c d\hat{\tilde{\lambda}}_{1\mu}^c}{2\pi} e^{i\hat{\tilde{\lambda}}_{1\mu}^c\left(\lambda_{1\mu}^c-\sum_{i=1}^{N}\frac{\tilde{W}_i^c(\tilde{x}_1)_i^\mu}{\sqrt{N}}\right)} \right.$$

$$\left. \times \int \prod_\mu \frac{du_{2\mu} d\hat{u}_{2\mu}}{2\pi} e^{i\hat{u}_{2\mu}\left(u_{2\mu}-\sum_{i=1}^{\rho N}\frac{(x_2)_i^\mu}{\sqrt{N}}\right)} \int \prod_{\mu,a} \frac{d\lambda_{2\mu}^a d\hat{\lambda}_{2\mu}^a}{2\pi} e^{i\hat{\lambda}_{2\mu}^a\left(\lambda_{2\mu}^a-\sum_{i=1}^{N}\frac{W_i^a(x_2)_i^\mu}{\sqrt{N}}\right)} \right\rangle_{\{\boldsymbol{x}^\mu\}}$$

$$\times \prod_{\mu,c} e^{-\frac{\beta}{2}\ell\left(\text{sign}\left(\tilde{u}_{1\mu}\right),\sigma\left(\tilde{\lambda}_{1\mu}^c\right)\right)} \prod_{\mu,a} e^{-\frac{\beta}{2}\ell\left(\text{sign}\left(u_{2\mu}\right),\sigma\left(\lambda_{2\mu}^a\right)\right)}.$$

Thus, the disorder average is now factorised and only involves exponential terms. Since the two datasets are independent now that we made the teacher explicit, we can take the averages over each one separately. In both cases we get:

$$\langle . \rangle = \prod_{i=1}^{\rho N} \mathbb{E}_{(x_{rel})_i^\mu} e^{-i\left(\frac{\hat{u}}{\sqrt{N}}+\sum_a \hat{\lambda}_a^\mu \frac{W_i^a}{\sqrt{N}}\right)(x_{rel})_i^\mu} \prod_{i=\rho N+1}^{N} \mathbb{E}_{(x_{irr})_i^\mu} e^{-i\left(\sum_a \hat{\lambda}_a^\mu \frac{W_i^a}{\sqrt{N}}\right)(x_{irr})_i^\mu}$$

$$= \prod_{i=1}^{\rho N}\left(1 - i\left(\frac{\hat{u}}{\sqrt{N}}+\sum_a \hat{\lambda}_a^\mu \frac{W_i^a}{\sqrt{N}}\right)\overline{x_{rel}} - \frac{1}{2}\left(\frac{\hat{u}}{\sqrt{N}}+\sum_a \hat{\lambda}_a^\mu \frac{W_i^a}{\sqrt{N}}\right)^2 Var\left(x_{rel}\right)\right)$$

$$\times \prod_{i=\rho N+1}^{N}\left(1 - i\sum_a \hat{\lambda}_a^\mu \frac{W_i^a}{\sqrt{N}}\overline{x_{irr}} - \frac{1}{2}\left(\sum_a \hat{\lambda}_a^\mu \frac{W_i^a}{\sqrt{N}}\right)^2 Var(x_{irr})\right) \quad \text{(B.5)}$$

$$= \prod_{i=1}^{\rho N}\left(1 - \frac{1}{2N}(\hat{u}^\mu)^2 - \frac{1}{N}\sum_a \hat{u}^\mu \hat{\lambda}_a^\mu W_i^a - \frac{1}{2N}\sum_{ab}\hat{\lambda}_a^\mu \hat{\lambda}_b^\mu W_i^a W_i^b\right)\prod_{i=\rho N+1}^{N}\left(1 - \frac{\Delta^\mu}{2N}\sum_{ab}\hat{\lambda}_a^\mu \hat{\lambda}_b^\mu W_i^a W_i^b\right)$$

$$= e^{-\frac{1}{2}\sum_{ab}\hat{\lambda}_a^\mu \hat{\lambda}_b^\mu\left(\frac{\sum_{i=1}^{\rho N}W_i^a W_i^b}{N}+\Delta\frac{\sum_{i=\rho N+1}^{N}W_i^a W_i^b}{N}\right)-\frac{\rho}{2}(\hat{u}^\mu)^2-\hat{u}^\mu \sum_a \hat{\lambda}_a^\mu \frac{\sum_{i=1}^{\eta N}W_i^a}{N}}. \quad \text{(B.6)}$$

This expression suggests what are the order parameters that capture the interactions of the model, namely:

- the teacher-student overlap at the end of the first learning phase: $\tilde{R}^c = \frac{\sum_{i=1}^{\rho N}\tilde{W}_i^c}{N}$.

- the teacher-student overlap at the end of the second learning phase: $R^a = \frac{\sum_{i=1}^{\rho N}W_i^a}{N}$

- the norm of the student after the first stage, decomposed into relevant/irrelevant parts: $\tilde{Q}_r^{cd} = \frac{\sum_{i=1}^{\rho N}\tilde{W}_i^c \tilde{W}_i^d}{N}$, $\tilde{Q}_i^{cd} = \frac{\sum_{i=\rho N+1}^{N}\tilde{W}_i^c \tilde{W}_i^d}{N}$

- the norm of the student after the second stage, decomposed into relevant/irrelevant parts: $Q_r^{ab} = \frac{\sum_{i=1}^{\rho N}W_i^a W_i^b}{N}$, $Q_i^{ab} = \frac{\sum_{i=\rho N+1}^{N}W_i^a W_i^b}{N}$

Therefore, after introducing these definitions by means of Dirac's $\delta$-functions, we can rewrite our replicated expression as:

$$\Omega^n = \int \prod_c \frac{d\tilde{R}^c d\hat{\tilde{R}}^c}{2\pi/N} \int \prod_a \frac{dR^a d\hat{R}^a}{2\pi/N} \int \prod_{cd} \frac{d\tilde{Q}_r^{cd} d\hat{\tilde{Q}}_r^{cd}}{2\pi/N} \int \prod_{cd} \frac{d\tilde{Q}_i^{cd} d\hat{\tilde{Q}}_i^{cd}}{2\pi/N} \int \prod_{ab} \frac{dQ_r^{ab} d\hat{Q}_r^{ab}}{2\pi/N} \int \prod_{ab} \frac{dQ_i^{ab} d\hat{Q}_i^{ab}}{2\pi/N}$$

$$\times G_i\, G_S\left(\hat{\tilde{R}},\hat{R},\hat{\tilde{Q}}_r,\hat{Q}_r\right)^{\rho N} G_S\left(0,0,\tilde{Q}_i,Q_i\right)^{(1-\rho)N} G_E\left(\Delta_1,\tilde{Q}_r,\tilde{Q}_i,\tilde{R},\tilde{n}\right)^{\alpha_1 N} G_E\left(\Delta_2,Q_r,Q_i,R,n\right)^{\alpha_2 N}$$
$$\text{(B.7)}$$

Where we introduced interaction, entropic and energetic potentials:

$$G_i = \exp\left(-N\left(\sum_c \hat{\tilde{m}}^c \tilde{m}^c + \sum_a \hat{m}^a m^a + \sum_{cd}\hat{\tilde{Q}}_r^{cd}\tilde{Q}_r^{cd} + \sum_{cd}\hat{\tilde{Q}}_i^{cd}\tilde{Q}_i^{cd} + \sum_{ab}\hat{Q}_r^{ab}Q_r^{ab} + \sum_{ab}\hat{Q}_i^{ab}Q_i^{ab}\right)\right)$$
$$\text{(B.8)}$$

$$G_S\left(\tilde{R},R,\tilde{Q},Q\right) = \int \prod_c\left[d\tilde{W}^c e^{-\frac{\beta\gamma}{2}(\tilde{W}^c)^2}\right] e^{-\frac{n\beta\gamma_{12}}{2}(\tilde{W}^1)^2} \int \prod_a\left[dW^a e^{-\frac{\beta(\gamma+\gamma_{12})}{2}(W^a)^2}\right]$$
$$\text{(B.9)}$$

$$\times \exp\left( \sum_c \hat{\tilde{R}}^c \tilde{W}^c + \sum_a \hat{R}^a W^a + \sum_{cd} \hat{\tilde{Q}}^{cd} \tilde{W}^c \tilde{W}^d + \sum_{ab} \hat{Q}^{ab} W^a W^b + \beta \gamma_{12} W^a \tilde{W}^1 \right)$$

$$G_E\left( \Delta, Q_r, Q_i, m, n \right) = \int \frac{du d\hat{u}}{2\pi} e^{iu\hat{u}} e^{-\frac{\rho}{2}(\hat{u})^2} \int \prod_{a=1}^n \frac{d\lambda^a d\hat{\lambda}^a}{2\pi} e^{i\lambda^a \hat{\lambda}^a} \qquad \text{(B.10)}$$

$$\times e^{-\frac{1}{2} \sum_{ab} \hat{\lambda}_a \hat{\lambda}_b \left( Q_r^{ab} + \Delta Q_i^{ab} \right) - \hat{u} \sum_a \hat{\lambda}_a R^a - \frac{\beta}{2} \ell(u, \lambda^a)}$$

**Replica Symmetric Ansatz**

The replica trick allowed us to express the average free-entropy as a function of the overlap order parameters. However, these objects are $n \times n$ matrices or $n$-dimensional vectors and in principle we have to average over all their possible realisations. Fortunately, the integrand function is exponential in $N$ and in the thermodynamic limit $N \to \infty$ the integrals are dominated by the extremisers of the action, and thus can be approximated with the saddle-point method. Still, we need a guess for how to parametrise these order parameters. The simplest possible ansatz, which turns out to be the correct one in convex problems as the one at hand, is the so-called Replica Symmetric ansatz, given by:

- $\tilde{R}^c = \tilde{R}$
- $R^a = R$
- $\tilde{Q}_{r/i}^{cd} = \tilde{q}_{r/i}$, for $c \neq d$; $\tilde{Q}_{r/i}^{cd} = \tilde{Q}_{r/i}$ for $c = d$.
- $Q_{r/n}^{ab} = q_{r/n}$ for $a \neq b$; $Q_{r/n}^{ab} = Q_{r/n}$ for $a = b$.

We also perform a Wick rotation $-i\hat{Q}_{ac,bd} \to \hat{Q}_{ac,bd}$ in order to deal with real valued conjugate parameters and pose a similar ansatz for them. In the next paragraph we will compute the three terms separately, and finally put them together in the expression for the RS free-entropy.

**Interaction term**

We start by evaluating the interaction term, or better its normalised logarithm $g_i = \lim_{\tilde{n} \to 0} \log G_i / (nN)$:

$$g_i = -\lim_{\tilde{n} \to 0} \frac{1}{n} \left( \tilde{n} \hat{\tilde{R}} \tilde{R} + n \hat{R} R + \tilde{n} \left( \frac{\hat{\tilde{Q}}_r \tilde{Q}_r}{2} + \frac{\hat{\tilde{Q}}_i \tilde{Q}_i}{2} \right) + \frac{\tilde{n}(\tilde{n}-1)}{2} \left( \hat{\tilde{q}}_r \tilde{q}_r + \hat{\tilde{q}}_i \tilde{q}_i \right) \right.$$

$$\left. + n \left( \frac{\hat{Q}_r Q_r}{2} + \frac{\hat{Q}_i Q_i}{2} \right) + \frac{n(n-1)}{2} \left( \hat{q}_r q_r + \hat{q}_i q_i \right) \right) \qquad \text{(B.11)}$$

$$= -\left( \hat{R} R + \frac{\left( \hat{Q}_r Q_r + \hat{Q}_i Q_i \right)}{2} - \frac{1}{2} \left( \hat{q}_r q_r + \hat{q}_i q_i \right) \right) \qquad \text{(B.12)}$$

In order to recover the optimisation problems entailed in the curriculum procedure, we now have to consider the zero temperature limit of this expression. When $\beta \to \infty$, the order parameters follow non-trivial scaling laws:

- $\hat{Q} \to \beta^2 \hat{Q} + \mathcal{O}(\beta), \hat{q} \to \beta^2 \hat{Q}$
- $(\hat{Q} - \hat{q}) \to -\beta \delta \hat{Q}$
- $\hat{R} \to \beta \hat{R}$
- $Q - q = \delta Q / \beta$

and similarly for the tilde parameters. Intuitively, looking at the last scaling law, we see that as the measure gets focused on the single minimiser of the loss, the overlap between different replicas $q$ rapidly converges to the norm $Q$. Moreover, the scaling with the inverse temperature of the conjugate

parameters prevents the interaction term from becoming sub-dominant in the saddle-point. If we substitute the rescaled parameters in the above expression we obtain:

$$g_i = -\beta \left( \hat{R}R + \frac{1}{2} \left( \hat{Q}_r \delta Q_r - \delta\hat{Q}_r Q_r \right) + \frac{1}{2} \left( \hat{Q}_i \delta Q_i - \delta\hat{Q}_i Q_i \right) \right) \tag{B.13}$$

**Entropic term**

We can now compute a similar quantity for the entropic potential, $g_i = \frac{\lim_{n \to 0}}{n} \log G_S \left( \tilde{R}, R, \tilde{Q}, Q \right)$. The general expression we will obtain can be specialised to the two cases $\left( \left\{ \tilde{R}, R, \tilde{Q}_r, Q_r \right\}, \left\{ 0, 0, \tilde{Q}_i, Q_i \right\} \right)$ appearing in the free-entropy. After substituting the RS ansatz we find:

$$
\begin{aligned}
g_S &= \lim_{\tilde{n} \to 0} \frac{1}{n} \log \int \prod_c \left[ d\tilde{W}^c e^{-\frac{\beta\gamma}{2}(\tilde{W}^c)^2} \right] e^{-\frac{n\beta\gamma_{12}}{2}(\tilde{W}^1)^2} \int \prod_a \left[ dW^a e^{-\frac{\beta(\gamma+\gamma_{12})}{2}(W^a)^2} \right] \tag{B.14}\\
&\quad \times \exp\left( \hat{\tilde{R}} \sum_c \tilde{W}^c + \hat{R} \sum_a W^a + \frac{1}{2}\left(\hat{\tilde{Q}} - \hat{\tilde{q}}\right) \sum_c \left(\tilde{W}^c\right)^2 + \frac{\hat{\tilde{q}}}{2}\left(\sum_c \tilde{W}^c\right)^2 + \right.\\
&\quad \left. + \frac{1}{2}\left(\hat{Q} - \hat{q}\right) \sum_a (W^a)^2 + \frac{\hat{q}}{2}\left(\sum_a W^a\right)^2 + \beta\gamma_{12} \sum_a \tilde{W}^1 W^a \right)\\
&= \lim_{\tilde{n} \to 0} \frac{1}{n} \log \int \mathcal{D}z \int \mathcal{D}\tilde{z} \int \prod_c \left[ d\tilde{W}^c e^{-\frac{\beta\gamma}{2}(\tilde{W}^c)^2} \right] e^{-\frac{n\beta\gamma_{12}}{2}(\tilde{W}^1)^2} \int \prod_a dW^a e^{-\frac{\beta(\gamma+\gamma_{12})}{2}(W^a)^2}\\
&\quad \times \exp\left( \frac{1}{2}\left(\hat{\tilde{Q}} - \hat{\tilde{q}}\right) \sum_c \left(\tilde{W}^c\right)^2 + \frac{1}{2}\left(\hat{Q} - \hat{q}\right) \sum_a (W^a)^2 + \right.\\
&\quad \left. + \left( \hat{\tilde{R}} + \sqrt{\hat{\tilde{q}}}\tilde{z} \right) \sum_c \tilde{W}^c + \left( \hat{R} + \beta\gamma_{12}W_1 + \sqrt{\hat{q}}z \right) \sum_a W^a \right)
\end{aligned}
$$

$$= \int \mathcal{D}z \int \mathcal{D}\tilde{z} \frac{\int d\tilde{W}e^{-\frac{1}{2}(\beta\tilde{\gamma} - (\hat{\tilde{Q}} - \hat{\tilde{q}}))\tilde{W}^2 + (\hat{\tilde{R}} + \sqrt{\hat{\tilde{q}}}\tilde{z})\tilde{W}} \log\left( \int dW \, e^{-\frac{1}{2}\left(\beta(\gamma+\gamma_{12}) - (\hat{Q} - \hat{q})\right)W^2 + \left(\hat{R} + \beta\gamma_{12}W_1 + \sqrt{\hat{q}}z\right)W} \right)}{\int d\tilde{W}e^{-\frac{1}{2}(\beta\tilde{\gamma} - (\hat{\tilde{Q}} - \hat{\tilde{q}}))\tilde{W}^2 + (\hat{\tilde{R}} + \sqrt{\hat{\tilde{q}}}\tilde{z})\tilde{W}}}$$

$$\tag{B.15}$$

In the zero-temperature limit, we consider the same rescaling of the order parameters we described above. The integrals over the weights become an extremum operation:

$$g_s = \lim_{\beta \to \infty} \beta \int \mathcal{D}z \int \mathcal{D}\tilde{z} M_s^\star, \tag{B.16}$$

where:

$$M_s^\star = \max_W \left\{ -\frac{1}{2}\left( (\gamma + \gamma_{12}) + \delta\hat{Q} \right) W^2 + \left( \hat{R} + \gamma_{12}\tilde{W}^\star + \sqrt{\hat{Q}}z \right) W \right\} \tag{B.17}$$

$$= \frac{1}{2} \frac{\left( \hat{R} + \gamma_{12}\tilde{W}^\star + \sqrt{\hat{Q}}z \right)^2}{(\gamma + \gamma_{12}) + \delta\hat{Q}} \tag{B.18}$$

and where: $\tilde{W}^\star = \text{argmax}_{\tilde{W}} \left\{ -\frac{1}{2}(\tilde{\gamma} + \delta\hat{\tilde{Q}})\tilde{W}^2 + (\hat{\tilde{R}} + \sqrt{\hat{\tilde{Q}}}\tilde{z})\tilde{W} \right\} = \frac{\hat{\tilde{R}} + \sqrt{\hat{\tilde{Q}}}\tilde{z}}{\tilde{\gamma} + \delta\hat{\tilde{Q}}}$.

Finally also the $\int \mathcal{D}z \int \mathcal{D}\tilde{z}$ integrations can be carried out, giving:

$$\beta \int \mathcal{D}z \int \mathcal{D}\tilde{z} M_s^\star = \beta \int \mathcal{D}z \int \mathcal{D}\tilde{z} \frac{1}{2} \frac{\left( \hat{R} + \gamma_{12}\frac{\hat{\tilde{R}} + \sqrt{\hat{\tilde{Q}}}\tilde{z}}{\tilde{\gamma} + \delta\hat{\tilde{Q}}} + \sqrt{\hat{Q}}z \right)^2}{(\gamma + \gamma_{12}) + \delta\hat{Q}} \tag{B.19}$$

$$= \frac{\beta}{2} \frac{\left(\hat{R} + \hat{\tilde{R}}\frac{\gamma_{12}}{\tilde{\gamma}+\delta\hat{\tilde{Q}}}\right)^2 + \left(\frac{\gamma_{12}\sqrt{\hat{\tilde{Q}}}}{\tilde{\gamma}+\delta\hat{\tilde{Q}}}\right)^2 + \hat{Q}}{(\gamma + \gamma_{12}) + \delta\hat{Q}} \tag{B.20}$$

So, specialising to the the two terms that appear in the free-entropy we get:

$$g_S(\gamma_1, \gamma_2, \gamma_{12}) = \rho\, g_s\left(\tilde{R}, R, \tilde{Q}_r, Q_r\right) + (1-\rho)\, g_s\left(0, 0, \tilde{Q}_i, Q_i\right) \tag{B.21}$$

$$= \frac{\beta}{2}\left(\rho \frac{\left(\hat{R} + \hat{\tilde{R}}\frac{\gamma_{12}}{\gamma_1+\delta\hat{\tilde{Q}}_r}\right)^2 + \left(\frac{\gamma_{12}\sqrt{\hat{\tilde{Q}}_r}}{\gamma_1+\delta\hat{\tilde{Q}}_r}\right)^2 + \hat{Q}_r}{(\gamma_2 + \gamma_{12}) + \delta\hat{Q}_r} + (1-\rho) \frac{\left(\frac{\gamma_{12}\sqrt{\hat{\tilde{Q}}_i}}{\gamma_1+\delta\hat{\tilde{Q}}_i}\right)^2 + \hat{Q}_i}{(\gamma_2 + \gamma_{12}) + \delta\hat{Q}_i}\right)$$

**Energetic term**

Since one of the two energetic terms appearing in the replicated free-energy depends on the $\tilde{n}$ replicas of the first weight configuration, and there is no interaction, we can take the $\tilde{n} \to 0$ limit directly. Therefore we only have to evaluate the other contribution (dependent on the $n$ replicas of the second weight configuration). Defining $Q = Q_r + \Delta Q_i$, $Q = Q_r + \Delta Q_i$, we evaluate $g_E = \lim_{n\to 0} \frac{1}{n} \log(G_E)$ in the RS ansatz:

$$g_E = \lim_{n\to 0} \frac{1}{n} \log \int \frac{du\, d\hat{u}}{2\pi} e^{iu\hat{u}} e^{-\frac{\rho}{2}(\hat{u})^2} \int \prod_a \left(\frac{d\lambda^a d\hat{\lambda}^a}{2\pi} e^{i\lambda^a\hat{\lambda}^a}\right) \tag{B.22}$$

$$\times e^{-\frac{1}{2}(Q-q)\sum_a\left(\hat{\lambda}_a\right)^2 - \frac{1}{2}q\left(\sum_a \hat{\lambda}_a\right)^2 - \hat{u}R\sum_a \hat{\lambda}_a - \beta\sum_a \ell(u,\lambda^a)}$$

$$= \lim_{n\to 0} \frac{1}{n} \log \int \frac{du}{\sqrt{2\pi\rho}} \int \prod_a \left(\frac{d\lambda^a d\hat{\lambda}^a}{2\pi} e^{i\lambda^a\hat{\lambda}^a}\right) \tag{B.23}$$

$$\times e^{-\frac{1}{2}(Q-q)\sum_a\left(\hat{\lambda}_a\right)^2 - \frac{1}{2}q\left(\sum_a \hat{\lambda}_a\right)^2 - \beta\sum_a \ell(u,\lambda^a) - \frac{1}{2\rho}\left(u + i R \sum_a \hat{\lambda}_a\right)^2}$$

$$= \lim_{n\to 0} \frac{1}{n} \log \int \mathcal{D}z_0 \int \mathcal{D}u \left\{\int \mathcal{D}\lambda\, e^{-\beta\ell\left(\sqrt{\rho}\,u, \sigma\left(\sqrt{(Q-q)}\lambda + \sqrt{q - \frac{m^2}{\rho}}z_0 + \frac{m}{\sqrt{\rho}}u\right)\right)}\right\}^n \tag{B.24}$$

$$= \int \mathcal{D}z_0 \int \mathcal{D}u \log \int \mathcal{D}\lambda\, e^{-\beta\ell\left(\sqrt{\rho}\,u, \sigma\left(\sqrt{Q-q}\lambda + \sqrt{q - \frac{m^2}{\rho}}z_0 + \frac{m}{\sqrt{\rho}}u\right)\right)}$$

So in the $\beta \to \infty$ limit, with the proper rescalings, we get:

$$g_E = \beta \int \mathcal{D}z \int \mathcal{D}u\, M_E^\star, \tag{B.25}$$

where:

$$M_E^\star = \max_\lambda -\frac{\lambda^2}{2} - \ell\left(\text{sign}\left(\sqrt{\rho}\,u\right), \sigma\left(\sqrt{\delta Q_r + \Delta\delta Q_i}\lambda + \sqrt{Q_r + \Delta Q_i - \frac{R^2}{\rho}}z + \frac{R}{\sqrt{\rho}}u\right)\right) \tag{B.26}$$

**RS Free-entropy**

Finally, assuming the we can write down the RS free-entropy for the curriculum ordering as:

$$\Phi/\beta = -\text{extr}\left(\hat{R}R + \frac{1}{2}\left(\left(\hat{Q}\delta Q - \delta\hat{Q}Q\right)_r + \left(\hat{Q}\delta Q - \delta\hat{Q}Q\right)_i\right)\right) \tag{B.27}$$

$$+ g_S(\gamma_1, \gamma_2, \gamma_{12}) + \alpha_2\, g_E\left(\Delta_2\right),$$

where $g_S$ is defined in equation (B.21) and $g_E$ is defined in equation (B.25). The order parameters for the teacher system are obtained independently from identical equations, after substituting $\lambda_1 \to 0, \lambda_2 = \to \lambda_1$ and $\lambda_{12} \to 0$, $\alpha_2 \to \alpha_1$ and $\Delta_2 \to \Delta_1$, and after adding a tilde to the remaining parameters.

The saddle-point equations, yielding at convergence the asymptotic prediction for the order parameters, can be found by posing stationarity conditions for the free-entropy with respect to all overlaps.

Note that, if instead of the simple setting just considered, where the data slice in the second stage has homogeneous variance for the irrelevant components, there are multiple subsets with different sizes and variances, the only variation in the free-entropy is in the energetic contribution. In general one will have a sum:

$$\sum_s \alpha_s g_E(\Delta_s) \tag{B.28}$$

over each of these subsets.

Moreover, if instead of two stages we consider multiple learning stages, the free-entropy for each successive step has an identical form, and one only has to substitute the tilde parameters with the order parameters obtained at the previous step. Note that the simplicity of nesting stages in this problem is connected to the convexity of this learning setting. Generally, adding more steps would increase the complexity of the calculation considerably.

**Generalisation error**

With the saddle-point values for the order parameters, one can easily evaluate the generalisation error on new datapoints, which is the measure of performance we are employing in the main. This performance can be obtained as:

$$1 - \epsilon_g = \left\langle \Theta\left(\left(\frac{\boldsymbol{W}_T \cdot x}{\sqrt{N}}\right)\left(\frac{\boldsymbol{W}_2 \cdot x}{\sqrt{N}}\right)\right)\right\rangle_{x(\Delta)} \tag{B.29}$$

where $\Delta$ is the variance of the irrelevant components for the new pattern. A shortcut for evaluating this expression is to insert the order parameters in the expression through Dirac's $\delta$s. After a straightforward calculation, along the same lines of the one presented above, one obtains:

$$\epsilon_g = \frac{1}{\pi}\arccos\left(\frac{R}{\sqrt{\rho\left(Q_r + \Delta Q_i\right)}}\right). \tag{B.30}$$

Of course, the generalisation accuracy is just the complementary quantity $1 - \epsilon_g$.

## C   Additional results on sparsity

We complement the discussion on the importance of sparsity, Sec. 4, with the comparison with other learning protocols. Observe that anti-curriculum suffers the same issue of the curriculum method for sufficiently large fractions of relevant features $\rho$. In that regime, the splitting becomes sub-optimal because the solution found in the splitting does not provide enough information to help the other phase of learning. Consequently, the network is forced to set neglect the information in the batch in favour of exploring solutions further away from that one. This is outperform by standard learning, where all the bits of information are used.

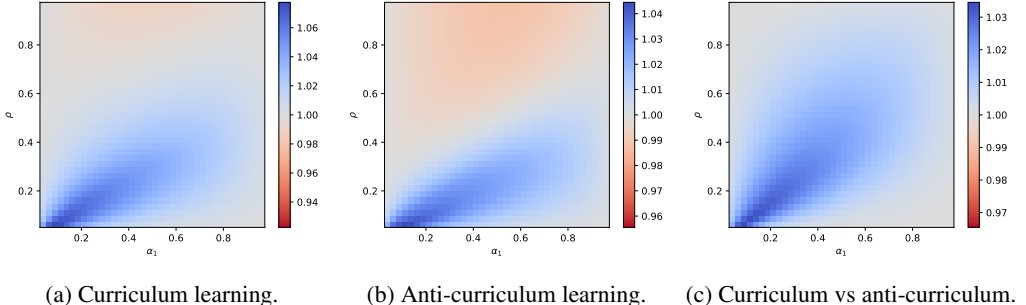

| (a) Curriculum learning. | (b) Anti-curriculum learning. | (c) Curriculum vs anti-curriculum. |

Figure C.1: **Effect of sparsity.** Phase diagram on the effect of sparsity, Fig. 4b, extended for all learning protocols.

## D    Simulations on CIFAR10

**Task design.** Because a sparse set of relevant features is crucial to observing curriculum effects in our model, we created a task based on real data that has this property. In particular we create $32 \times 64$ pixel input examples by concatenating two images side-by-side from the CIFAR10 dataset. The correct output label is given by the label of the image on the left, while the image on the right is an irrelevant distractor. To vary difficulty, we scale the contrast of the irrelevant image. This dataset is meant to instantiate a simple example of learning an object classification amidst clutter. We emphasise that, as in our synthetic data model, each training sample always contains the same relevant and distractor images (i.e., we are not considering a data augmentation setting where each relevant image appears with many non-relevant images). To ensure no cross-contamination of training and testing samples, the distractor images for the training and test sets are drawn only from the same set.

**Model architecture and training regime.** We train a single layer network with cross entropy loss (i.e. softmax regression), implemented in Pytorch Lightning by modifying the MIT-licensed `PyTorch_CIFAR10` repository (`https://zenodo.org/record/4431043#.YLmz6zZKhsA`) to ensure that training parameters accord with standard practice. Networks were trained with SGD and Nesterov momentum, under default parameters: a learning rate of $1e - 2$, momentum parameter 0.9, batch size 256, and 100 epochs. The learning rate was annealed according to the 'WarmUpCosine' schedule used in `PyTorch_CIFAR10`, which linearly reduces the learning rate over the first $30\%$ of training steps before switching to a cosine shaped schedule on the remainder.

**Experiment details and hyperparameter optimisation.** For the first phase of training, we used dataset sizes in 10 equal steps between 1000 and 50000. For the second phase, we used nine dataset sizes in 9 equal steps between 5333 and 48000. We optimised hyperparameters in each phase separately. In the first phase, we evaluated all combinations of initialisation scales of $\{0, .2, .5, 1.\}$, weight decay parameters of $\{0, .2, .5, 1., 2.\}$, and curriculum policy, for five random seeds. In the second phase, for each random seed and curriculum condition, we continued training from the best-performing model obtained in the first phase. We trained all combinations of five elastic penalties log spaced between $1e - 3$ and $1e2$, and weight decay parameters $\{0, .2, .5\}$. We then compute the best performing model for each seed and take the mean over seeds. Finally, to evaluate the no-curriculum performance, we train shuffled dataset models with initialisation scales $\{0, .2, .5, 1.\}$ and weight decay parameters $\{0, .2, .5\}$. For visualisation purposes, we used nearest-neighbors interpolation in the phase portrait to provide values for all points used in the synthetic experiments. Experiments were run on V100 GPUs and required approximately 10000 GPU hours (including debugging and development), or $\approx 1110$ kg $CO_2$ eq according to the MachineLearning Impact calculator of Lacoste et al., 2019.

## E    Speed-up theory vs simulations

As remarked in the main text, one of the advantages of the theoretical analysis is a huge speed-up in the time to collect the results, without need of averaging to reduce the fluctuations. In this section, we briefly report a comparison between the time required for the lines from theory and simulations shown in the main text.

In order to obtain figure 1c, a single run of the ODE equations takes 2 milliseconds and a run of the simulations takes 500 milliseconds. The figure is however obtained optimizing over all the hyperparameters (learning rate, initialization, weight decay) totalling 400 milliseconds for the analytical solution; while, due to noise, simulation results for a single set of hyperparameters requires averaging 5000 realizations totalling 41 minutes. We note that we did the hyperparameter optimization only once using the theoretical framework and then used the optima in the simulations in order to save compute time. The best comparison should therefore be done for a fixed set of hyperparameters and gives 2 milliseconds vs 41 minutes. Overall, the analytical solution is between 2 and 6 orders of magnitude faster.