# OpenReview forum: "An Analytical Theory of Curriculum Learning in Teacher-Student Networks"
_NeurIPS.cc/2022/Conference — NeurIPS 2022 Accept_

### Official Review · Reviewer_MUSa · 2022-07-08

**Rating:** 5
**Confidence:** 3
**Soundness:** 4 excellent
**Presentation:** 1 poor
**Contribution:** 3 good

**Summary:**

This work introduces an analytical framework that eases the study of the effect of data curricula in teach-student networks. The authors present a set of formulas that can produce the final accuracy of a model based on the initial parameters of the student and teacher networks without the need to perform the simulation (training). The analytical solution is an iterative method that is applied to the online learning case (optimize on one sample at a time, and each sample is used only once) and batch learning case (train until convergence). They evaluate their methods and study the benefits of curricula using a synthetic dataset of Gaussian samples whose label is determined by a sparse teacher network and difficulty is determined by the variance in irrelevant samples. Additionally, they evaluate on a custom-designed task using CIFAR-10 data, where an irrelevant image is concatenated to the target image, and difficulty is based on the opacity of the irrelevant image. The study the effect of curricula based on model initialization scale, teacher network sparsity level, and variance amount in irrelevant features (SNR).

**Questions:**

- *"due to the convex nature of the teacher-student setup [22], the network is bound to converge to a minimum uniquely determined by the final slice of data, with no memory of the progress made at intermediate steps"* ... Is this a property of all student-teacher networks or just the the setup you are using? Because if it isn't, then curriculum learning can have a significant effect to the learning trajectory in the parameter spaces, and the optimum points able to be reached.

- *"the first stage in the curriculum strategy can only help in the support identification problem"* ... Shouldn't the identification of the support lead to easy, successful learning of the simple remaining task?

- What do you mean by training trajectories? Is it the accuracy curve over the course of training?

**Limitations:**

- I am not sure how generalization the teacher-student network analysis is, as compared to standard learning approaches. In particular, this setup with a convex network, simple optimizer, and simple synthetic dataset. I think the learning dynamics can be significantly different in a deep, non-convex network and using a modern momentum-based optimizer.

**Strengths And Weaknesses:**

**Strengths**

- An analytical theorem, with a rigorous proof, which can be of great use to the study of curricula and learning theory.
- The connection of machine learning results with experiments of psychology is interesting, and convincing, and attests to the soundness of the author's methods.
- Well-written introduction about curriculum learning with references to biological, psychological and artificial learning backgrounds.
- Investigates an important problem and presents a solid problem definition, particularly in the batch learning section.

**Weaknesses**

- The benefits of the analytical solution are states as a) "free of finite size effects and stochastic fluctuation", and b) "their evaluation is very fast". Both of them require further elaboration. I would like to see a comparison of the speed of the analytical solution and full training, and a break-down of how the analytical solution is faster. In the supplementary material I see that the analytical forms of the updates are computed using matrix multiplications and activation functions on a scale comparable to the student network's computations.
- Section two is title "Online dynamical solution in the large input limit", but then the relevance of the large input limit (or even its definition) is not explained. Furthermore, the relevance and effect of high-dimensional input is not clearly shown.
- Between the analytical theory, experiments on the benefit of CL, introduction of elastic coupling method, and connections between machine learning and psychology, the contributions of the paper seem scattered and unfocused.
- The organization and writing throughout the paper is quite confusing.
  - Figure 1 is based on notations and definitions introduced in section 3 and is not easily understood at the point it is presented.
  - Line 137: "Fig. 5 shows performance as a function of sparsity ρ". Figure 5 does not show that. I think you referenced the wrong figure.
  - Line 140: "low-D tasks .. [6]". The notation low-D is not introduced, not even in [6]. I assumed it means low-dimensional.
  - Line 143: "starting from a large initialisation", I assume it means the random initialization scale of the student or teacher network. It is not entirely clear. Please include more details about this initialization. Does it use a normal random distribution?
  - Figure 1: "The curriculum boundary lies at α = 1/2". What is the curriculum boundary? It is never defined. The abstract talks about "curriculum boundary consolidation", but it is not further elaborated.
  - What is α in Fig 1c x-axis and other figures? $\alpha_1$ and $\alpha_2$ are defined, but $\alpha$ never is, and I am not able to understand what it means.
  - Half-way through the paper section numbers stop being used. What I assume are sections 5 and 6 do not have numbers. And I am not sure if the subsections of Section 4 belong there.
  - Figure 3c "Accuracy hard samples." is not an accurate title. It is accuracy of all samples, easy and hard. It's just that anti-curriculum performs best, which is influenced by hard samples.
  - The figure captions are inconsistent in the uses of (a), (b), (left, center, right), and (top, bottom). For example Figure 3 says (a), (b) and then "the right panel".

---

> ### Author Response · Authors · 2022-07-29
> **Reply 1/2**
>
> We thank the reviewer for the detailed comments. We are glad the reviewer appreciated the soundness of our theory, the relevance of connecting experimental psychology with ML theory, and the significance of the problem.
>
> * *“I would like to see a comparison of the speed of the analytical solution and full training, and a break-down of how the analytical solution is faster”* **Reply** In order to obtain figure 1c, a single run of the ODE equations takes ~2 milliseconds and a run of the simulations takes ~500 milliseconds. The figure is however obtained optimizing over all the hyperparameters (learning rate, initialization, weight decay) totalling ~400 milliseconds for the analytical solution; while, due to noise, simulation results for a single set of hyperparameters requires averaging 5000 realizations totalling ~41 minutes. We note that we did the hyperparameter optimization only once using the theoretical framework and then used the optima in the simulations in order to save compute time. The best comparison should therefore be done for a fixed set of hyperparameters and gives ~2 milliseconds vs ~41 minutes. It is true that both operations involve matrix multiplications, but in the case of the simulations they use the actual input dimension N (=500 for figure 1) and in the case of the theory the effective dimension is 4 (i.e. the number of order parameters). Overall, the analytical solution is between 2 and 6 orders of magnitude faster. We will add this approximate speed-up factor and discussion to the supplement (or if space allows, the revision).
> * *“the relevance of the large input limit (or even its definition) is not explained”* **Reply** The large input limit means that the input size N and the dataset size M go to infinity with finite ratio \alpha=M/N. This is an important point that must have gotten lost during the iterations, thank you for catching this. We will reintroduce it in the revised version.
> * *“the contributions of the paper seem scattered and unfocused”* **Reply** The revised version will have a bullet point list with the main contributions at the end of the introduction. The aim of this work is to develop a unifying theory that is meant to make phenomena in both ML and Psych look similar and more transparent.
> * *“The organization and writing throughout the paper is quite confusing.”*
> * * *Figure 1 is based on notations and definitions introduced in section 3 and is not easily understood at the point it is presented.* **Reply** We will replace the image with explicit notation.
> * * *Line 137: "Fig. 5 shows performance as a function of sparsity ρ". Figure 5 does not show that. I think you referenced the wrong figure.* **Reply** Yes, that was supposed to point to Fig.4b, thank you for the careful read.
> * * *Line 140: "low-D tasks .. [6]". The notation low-D is not introduced, not even in [6]. I assumed it means low-dimensional.* **Reply** Yes, we meant low-dimensional.
> * * *Line 143: "starting from a large initialisation", I assume it means the random initialization scale of the student or teacher network. It is not entirely clear. Please include more details about this initialization. Does it use a normal random distribution?* **Reply** Yes, we use a normal distribution with a fixed variance. When we refer to large/small initialization we mean large/small initial variance. We will clarify this in the revised version.
> * * *Figure 1: "The curriculum boundary lies at α = 1/2". What is the curriculum boundary? It is never defined. The abstract talks about "curriculum boundary consolidation", but it is not further elaborated.* **Reply** We mean the switching point between the two levels of difficulty. We will clarify this.
> * * *What is α in Fig 1c x-axis and other figures?* **Reply** \alpha in the case of online learning is the rescaled time. In general \alpha represents (dataset size M)/(input dimension N), in the case of online learning the dataset size coincides with time since every step corresponds to a new sample added to the dataset.
> * * *Half-way through the paper section numbers stop being used. What I assume are sections 5 and 6 do not have numbers. And I am not sure if the subsections of Section 4 belong there.* **Reply** We will add 5 and 6 to the last two sections.
> * * *Figure 3c "Accuracy hard samples." is not an accurate title. It is accuracy of all samples, easy and hard. It's just that anti-curriculum performs best, which is influenced by hard samples.* **Reply** We will replace it with just accuracy.
> * * *The figure captions are inconsistent in the uses of (a), (b), (left, center, right), and (top, bottom). For example Figure 3 says (a), (b) and then "the right panel".* **Reply** We will use only the a,b,c notation.

---

> ### Author Response · Authors · 2022-07-29
> **Reply 2/2**
>
> * *"due to the convex nature of the teacher-student setup [22], the network is bound to converge to a minimum uniquely determined by the final slice of data, with no memory of the progress made at intermediate steps" ... Is this a property of all student-teacher networks or just the the setup you are using? Because if it isn't, then curriculum learning can have a significant effect to the learning trajectory in the parameter spaces, and the optimum points able to be reached.* **Reply** As stated in the paper, the answer does depend on the setup and in principle we would expect different behaviours for non-convex models. In particular, the presence of different basins of attraction towards different minima would suggest that initializing close to a good one would produce a performance improvement. However, note that empirical results in the ML field are not showing clear signals in this direction. A possible explanation is that relying on memory effects in the learning dynamics would require one to hit a sweet spot in the learning rate value and in the number of training epochs, and this seems hard to be achieved consistently. For this reason, we speculate that explicitly enforcing this memory by altering the loss with the curriculum information could be useful even in these settings. We will further emphasize that this statement about memorylessness applies only to our setting.
> * *"the first stage in the curriculum strategy can only help in the support identification problem" ... Shouldn't the identification of the support lead to easy, successful learning of the simple remaining task?* **Reply** It does, but when the remaining data is insufficient this will still be not enough to solve the problem. In particular, the sentence refers to figure 4. To clarify, after the support has been determined the effective sample complexity is rescaled by \rho (because effectively the optimization problem concerns \rho*N variables instead of N). So for fixed \alpha_1 and \alpha_2, in the curriculum setting high \rho requires more samples to achieve the same performance as a setting with less data but higher sparsity. In contrast, standard learning is not affected by this tradeoff and this explains the results of figure 4.
> * *What do you mean by training trajectories? Is it the accuracy curve over the course of training?* **Reply** Yes, but not only. We mean all the quantities that could be measured during learning including test error, overlaps, etc.
> * *I am not sure how generalization the teacher-student network analysis is, as compared to standard learning approaches.* **Reply** We are currently working to address some of these limitations: more complex network architectures and more complex data models. Each of these ingredients entails a large jump in the technical complexity of the analysis and requires a not trivial generalization of this work. We remark that most of the theoretical analyses on typical learning scenarios focus on convex setups. In the context of curriculum learning, even this simpler case had never been studied analytically previous to this work. We are aware of the limitations of our modelling setup, and that more complex models could be more representative of real-world behaviour, but jumping to these models without understanding the simpler ones would be very challenging. We hope our work, which solves a model of curriculum learning that was proposed independently, is a prerequisite to understanding more complex scenarios.
>
> We thank the reviewer for pointing out several elements that were missing in this version: we will make sure to include them in the revised version of this paper.

---

> > ### Comment · Reviewer_MUSa · 2022-08-05
> > **Response to Rebuttal**
> >
> > Thanks to the authors for their effort and clarifications.
> >
> > The setup based on Bengio's 2009 Gaussian toy experiment has limited consequences to modern deep neural networks. However, I recognize its value to learning theory and the development of curriculum learning.
> >
> > The authors say "*We hope our work, which solves a model of curriculum learning that was proposed independently, is a prerequisite to understanding more complex scenarios.*"
> > Would the authors provide a proposal on how the analysis of complex scenarios may benefit from this work?

---

> > > ### Author Response · Authors · 2022-08-07
> > > **Reply to the reviewer**
> > >
> > > We thank the reviewer for this stimulating question. We agree the road ahead is likely long, but in our view, the main hope of bringing theory up to speed with practice is to start with these simpler (but still involved, from the theory perspective) settings rather than trying to jump directly to full state-of-the-art settings. The analysis presented in this work can be helpful in different ways.
> > >
> > > First, the analyzed toy model is very elemental in nature but the curriculum phenomenology can already be observed with it. In settings with an increased degree of complexity, the imputation of correlation vs causation may become subtler: these preliminary results can help identify the true causes of curriculum effects and discern them amidst the various additional modelling elements.
> > > On the other hand, if increasing the complexity disrupts the effectiveness of curriculum, we can use our baseline model to highlight the relevant differences between the two learning setups.
> > >
> > > Second, this simple model provides us with a strong foundation for designing further modelling setups and experiments, and for producing informed hypotheses on the expected outcomes. For example, here's a list of observations that can help defining a roadmap for future work:
> > > * 1- We have confirmed the empirical observation that curriculum should speed up the learning dynamics, even in the most basic setup. We have also established that curriculum effects can be sizeable even in equilibrium analyses and that it is not just a dynamic effect. This means that looking at the equilibrium properties of more complex models (e.g. with the stat phys approach employed in this work and the technical tool of the Franz-Parisi computation) can be an interesting direction for future research.
> > > * 2- In our study, we have observed the importance of sparsity of signal and noise dimensions. Building from the sparsity direction, it may be possible to generalise the analysis to the hidden manifold model in which a small underlying subspace generates high dimensional observations through a nonlinear process. This model has been shown to capture additional features of realistic classification settings, in which an underlying manifold of images yields high-dimensional pixel observations.
> > > In general, our observation suggests that adding confounding effects and decreasing the signal-to-noise ratio can increase the effectiveness of curriculum.
> > > * 3- We also view our work as a prerequisite to extending the analysis toward 'nested' tasks in which certain simpler tasks are useful prerequisites for more complex tasks (classifying MNIST images might be useful for a system that solves simple arithmetic questions, for instance). Here this could involve 'nested' teacher networks and students, but the same principles and analysis methods would be expected to form an essential component.
> > > * 4- We have understood that the boundaries between curriculum stages are important and that a memory effect is needed in order to observe an improvement in learning performance. While in principle this effect should appear naturally in non-convex settings, the empirical observations in ML experiments have shown that exploiting these landscape effects could be non-trivial. The modified loss proposed in this paper could thus be useful even in non-convex scenarios, to help "localize" the learning dynamics and strengthen the memory effect. Moreover, this approach could be useful in the theoretical analysis to give direct control of the learning trajectory.
> > > * 5- We have seen that parameter initialization and regularization can be key for observing curriculum effects (in particular a small parameter initialization). This suggests a working hypothesis that overparametrization could diminish curriculum effects, and that the effectiveness of curriculum learning could emerge more naturally in constrained settings. This is an object of current investigations.
> > > * 6- In some cases obtaining entire phase diagrams could become more expensive. The results presented in this paper could help design targeted experiments, focusing the computational resources on the regions where interesting phenomena are more plausible, or where some effects are more likely to appear. This type of informed guesses can help explore the vast parametric landscapes of more complex models.
> > >
> > > Building from the idea of curriculum-dependent algorithms, we hope that analysis of more complex settings will suggest more sophisticated curriculum-dependent algorithms and shed light on curriculum design, which has become important in parts of state-of-the-art reinforcement learning systems. We remain motivated by the observation that curriculum is highly valuable for human and animal learning, suggesting that there should be settings where curriculum makes a large practical difference.

---

> > > > ### Comment · Reviewer_MUSa · 2022-08-08
> > > > **Review Summary**
> > > >
> > > > Thanks to the authors for answering my questions.
> > > >
> > > > I find the ideas presented in this paper valuable to the community, and beneficial for future research in curriculum learning.
> > > >
> > > > However, I find the quality of presentation at the time of submission inappropriate for a publication.
> > > >
> > > > At this point, I will raise my score by one point recommending a tendency to accept, contingent on the improvement of the presentation in a revised version.

---

### Official Review · Reviewer_S1ip · 2022-07-11

**Rating:** 6
**Confidence:** 4
**Soundness:** 3 good
**Presentation:** 3 good
**Contribution:** 2 fair

**Summary:**

The paper explores the mechanism of curriculum learning, specifically when will, or will not, the curriculum setting work. The paper investigates two kinds of demos, online learning dynamics, and batch learning, as well as proposes an elastic penalty regularization, which demonstrates that the curriculum setting could only be beneficial when the task is sparse. The paper also validates the observation on CIFAR10, which further illustrates the investigation over real datasets.

**Questions:**

1. In Figure 2, the paper presents that under some circumstances, anti-curriculum learning could perform better than curriculum or non-curriculum learning (b,c bottom left part), which is anti-intuitive, can the paper give some explanations of such phenomenon?
2. The paper reveals that with too less easy data ($\alpha_1$), the performance of curriculum learning can be degraded, can it be related to overfitting?



**Limitations:**

See Sec 'Strengths And Weaknesses'

In addition, the Appendix is not well related to the main text, please use markers in the main text to direct the appendix list.

**Strengths And Weaknesses:**

Strenght:
1. The paper is well organized, and clearly presented, in which it goes from easy to deep to introduce what curriculum learning is, its limitations, and investigate when curriculum settings will benefit or not.
2. The paper investigates the relations between curriculum learning and sparsity, and demonstrates that the effectiveness of curriculum learning heavily depends on the sparsity.

Weaknesses:
1. Though the paper introduces the experiments over CIFAR10, the evaluation over the real dataset is still somehow limited, it could be better to have more evaluations of curriculum learning on other real data with non-convex settings, or at least make the explanation of the evaluation on CIFAR10 more detailed.
2. It can be better (though not necessarily to have) if some methodologies could be proposed to improve the performance of curriculum learning under non-sparse cases, which could make it more practical.

---

> ### Author Response · Authors · 2022-07-29
> **Reply**
>
> We thank the reviewer for their time and comments. We are happy to read that the reviewer found our paper clearly written and appreciated the way we demonstrated the importance of sparse features in curriculum learning.
>
> * *“it could be better to have more evaluations of curriculum learning on other real data with non-convex settings, or at least make the explanation of the evaluation on CIFAR10 more detailed.”* **Reply** The first point raised by the reviewer is the object of current investigations (see answer below). We will add more details on the CIFAR experiments in the revised version.
> * *“if some methodologies could be proposed to improve the performance of curriculum learning under non-sparse cases”* **Reply** We introduced sparsity as the simplest form of data structure. In general, we know data spans a low-dimensional manifold and in our model the sparsity is used to mimic this aspect. A more realistic setting would still require a more complex model of the low-dimensional structure but it would be harder (or even impossible) to have an exact solution to the dynamics. A still solvable setting, but technically not straightforward, may be this one https://journals.aps.org/prx/abstract/10.1103/PhysRevX.10.041044. Understanding the connection between landscape complexity and curricula is, indeed, a direction that we are currently pursuing. We hope the present work, which solves this sparse setting proposed independently, represents a step along this longer path.
> * *“In Figure 2, the paper presents that under some circumstances, anti-curriculum learning could perform better than curriculum or non-curriculum learning (b,c bottom left part), which is anti-intuitive, can the paper give some explanations of such phenomenon?”* **Reply** We agree that this appears counter-intuitive. We also have hoped for greater intuition on this point, but we do not have a simple explanation for this phenomenon. This result is what appears from solving the equations and it was checked in the numerical simulations. A possible intuition could be that, in some settings, the large amount of noise contained in the hard data will always be too disruptive for effective learning. Thus, leaving the “clean” data for last could allow the model to better exploit the easy data. We will add this possibility to the revision. Even without a clear intuition, our contribution here is to show, in an identical setting and without finite size effects, that both anticurriculum and curriculum can indeed outperform the baseline.
> * *“The paper reveals that with too less easy data, the performance of curriculum learning can be degraded, can it be related to overfitting?”* **Reply** In some sense. The point of this simple model is that the irrelevant directions should be ignored by the learning model, even when their variance is large (in the hard samples); however, the model would naturally try to overfit these noise directions, causing a degradation of the performance. The interplay between better aligning to the signal directions and risking overfitting the noise is quite complicated. Indeed, in the scenario mentioned by the reviewer, curriculum is not performing badly, it is just out-performed by anti-curriculum. Even in this simple model, it is not completely clear why anti-curriculum is so effective in this setting (which connects to the previous question).
>
> We will make adjustments in the revised version in order to meet some of the proposed changes.

---

> > ### Comment · Reviewer_S1ip · 2022-08-06
> > **Response to Rebuttal**
> >
> > Thanks to the author's response and clarifications.
> >
> > I'd believe the rebuttal has addressed most of my concerns in the last review. Though this is more like a learning theory paper, I'd believe it's good to relate the theory with real data rather than some simple generated data/settings, and the investigation on CIFAR10 should be enough to verify their theory (though the more dataset, the better). The author claims 'We will add more details on the CIFAR experiments in the revised version, which can be beneficial to understand the experiments results and theory better than the current version.
> >
> > Also, the author gives possible intuitions towards the anti-normal phenomenons proposed, though some interpretations can be good enough for current work, I'd expect some toy experiments to verify the proposed claims in the rebuttal, which can make the paper more convincing and comprehensive.
> >
> > Overall, I think the rebuttal resolves my concerns and I'll keep my decision to recommend the tendency to accept.

---

### Official Review · Reviewer_Cgtb · 2022-07-12

**Rating:** 6
**Confidence:** 3
**Soundness:** 3 good
**Presentation:** 3 good
**Contribution:** 3 good

**Summary:**

The paper is motivated by the gap in our understanding of the importance of curriculum learning in biological models vs machine learning models. The main question the paper tries to address is, when and why curriculum learning helps.
They consider a teacher, student setup where both teacher and student are shallow 1-layer neural networks of size N and the teacher network is sparse, with only a fraction ρ < 1 of ∼ N (0, 1) non-zero components. The student model has to guess which components should be set to zero and align the relevant weights in the correct direction.
The findings of the paper are as follows:
* Curriculum learning is most effective when a small signal is embedded amidst many irrelevant feature.
* The effect of curriculum or anti-curriculum depends on other learning parameters such as initialisation (anti-curriculum can lead to better results of hyper-parameters are not optimal).
* Curriculum mainly offers a dynamical advantage: it speeds learning, with only minimal impact on asymptotic performance.
* Curriculum-aware algorithms can better exploit curricula.

**Questions:**

- One way to explain why it seems that explicit curriculum learning doesn't help in many cases in ML, is that the models could potentially be impacted by some imiplict curriculum, e.g., curriculum  imposed by the architecture of the model or the optimization algorithm. What do you think about this, and how does it fit to the dicussions in this paper?

- What does "curriculum-aware algorithms" mean?

**Limitations:**

I don't have any comments on this.

**Strengths And Weaknesses:**

The topic of the paper is very interesting and inspiring: linking psycological experiments and theories of how curriculum learning works/helps humans/animals to our understanding of effects and benefits of curriculum learning in machine learning.

The main issue I think is that the emprical experiments are too simple and far away from the state of the art ML techniques. I wonder how, conceptually, the findings could be replicated in scaled up settings.

---

> ### Author Response · Authors · 2022-07-29
> **Reply**
>
> We thank the reviewer for their comments and for acknowledging the relevance of the work in connecting behavioural psychology and ML theory.
>
> * *“the emprical experiments are too simple and far away from the state of the art ML techniques. I wonder how, conceptually, the findings could be replicated in scaled up settings.”* **Reply** In the paper, we refer to empirical studies employing SOTA algorithms and more complex benchmarks [e.g. https://arxiv.org/abs/2012.03107]. Our work attempts to put all the empirical findings together and to find an explanation based on our understanding of a simple solvable model. While SOTA methods keep evolving, we believe that a certain degree of universality is preserved and elements observed in simple models can be transferred to more complex scenarios. We validated our theory on CIFAR, and these numerical experiments show similar features to those observed in the theoretical framework.
> * *“One way to explain why it seems that explicit curriculum learning doesn't help in many cases in ML, is that the models could potentially be impacted by some imiplict curriculum, e.g., curriculum imposed by the architecture of the model or the optimization algorithm. What do you think about this, and how does it fit to the dicussions in this paper?”* **Reply** Unfortunately, implicit curricula cannot be directly studied within our framework, since we always assume the hardness information is completely disclosed. Considering a pseudo-labeling step before the actual learning stage would likely make the (already involved) computation unfeasible. Empirically this procedure has been investigated in [e.g. https://arxiv.org/abs/1812.05159, https://arxiv.org/abs/2012.03107] and, limiting our interest in the generalization aspect, this heuristic does not seem to induce a sizeable improvement. We will include this discussion in the revision.
> * *“What does "curriculum-aware algorithms" mean?”* **Reply** We mean an algorithm that explicitly depends on the curriculum, for instance by changing its objective function when example difficulty changes. That is, a curriculum aware algorithm would adapt the learning process in order to account for different levels of difficulties in the data. A simple way of implementing this is to modify the training loss, as proposed in this paper. Other approaches may involve adapting the optimization algorithm as proposed in https://proceedings.mlr.press/v139/ruiz-garcia21a.html, or possibly modifying the architecture https://www.sciencedirect.com/science/article/pii/0010027793900584. A key message emerging from our work is that standard algorithms do not dramatically benefit from curriculum, and we believe curriculum-aware algorithms may be the way forward. We will include a clear definition and discussion in the revision.
>
> We appreciate the comments provided by the reviewer, pointing out some issues in the presentation. We will clarify all these points in the revised version and try to better convey our message.

---

> > ### Comment · Reviewer_Cgtb · 2022-08-06
> > **Thanks a lot for your response.**
> >
> > Thanks a lot for the time you took to prepare the rebuttal and for the clarifications.
> >
> > I don't have anything else to add at this point. Just wanted to aknowledge that I read all the reviews and the rebuttal. I really appreciate author's efforts in preparing the rebuttal. I think the paper does a good job is solving a pieace of a puzzle of understanding curriculum learning in ML but I think (as pointed out by other reiewers) the presentation can be improved. At this point, I am standing by my initial score.
> >
> > My level of expertise and background doesn't allow me to judge the paper beyond this but I like the paper and would have to read it a few more times to be able to fully understand it.

---

### Meta-Review · Area_Chair_L9e3 · 2022-08-30

**Recommendation:** Accept
**Confidence:** Certain

**Metareview:**

In this paper, authors provide an analytical theory of curriculum learning for an online teacher-student setting where a subset of features are relevant and are used by the teacher while a student might get distracted and use irrelevant features. In such a setting, the difficulty of examples can be captured by the variance in the irrelevant features. The insights from this analysis help explain existing empirical observations reported in prior work for effectiveness of curriculum learning as opposed to random-ordering and anti-curriculum learning in compute-limited regime. Further interesting connections to the literature on cognitive science are discussed in the paper.

Given the lack of theoretical basis for curriculum learning, reviewers are in agreement that this paper is in-time and impactful for the ML community and they all recommended accepting the paper. However, during the discussion period it became clear that this recommendation was based on the promise of authors to revise the paper which never happened during the rebuttal period.

My final recommendation is to accept the paper based on the authors' promise that the final version will include all changes promised by authors in their response to reviewers. Although there is no formal conditional acceptance mechanism, the authors should view this as a conditional acceptance based on taking their word that they will revise accordingly (see the list of changes below). I will check the camera-ready version and call-out anything less than that.

**List of changes promised by authors (quoted from their response)**

**Authors' response to reviewer Cgtb:**

1- "Unfortunately, implicit curricula cannot be directly studied within our framework, since we always assume the hardness information is completely disclosed. Considering a pseudo-labeling step before the actual learning stage would likely make the (already involved) computation unfeasible. Empirically this procedure has been investigated in [e.g. https://arxiv.org/abs/1812.05159, https://arxiv.org/abs/2012.03107] and, limiting our interest in the generalization aspect, this heuristic does not seem to induce a sizeable improvement. We will include this discussion in the revision."
2- "We mean an algorithm that explicitly depends on the curriculum, for instance by changing its objective function when example difficulty changes. That is, a curriculum aware algorithm would adapt the learning process in order to account for different levels of difficulties in the data. A simple way of implementing this is to modify the training loss, as proposed in this paper. Other approaches may involve adapting the optimization algorithm as proposed in https://proceedings.mlr.press/v139/ruiz-garcia21a.html, or possibly modifying the architecture https://www.sciencedirect.com/science/article/pii/0010027793900584. A key message emerging from our work is that standard algorithms do not dramatically benefit from curriculum, and we believe curriculum-aware algorithms may be the way forward. We will include a clear definition and discussion in the revision."

**Authors' response to reviewer S1ip:**

1- "The first point raised by the reviewer is the object of current investigations (see answer below). We will add more details on the CIFAR experiments in the revised version."

2- "We agree that this appears counter-intuitive. We also have hoped for greater intuition on this point, but we do not have a simple explanation for this phenomenon. This result is what appears from solving the equations and it was checked in the numerical simulations. A possible intuition could be that, in some settings, the large amount of noise contained in the hard data will always be too disruptive for effective learning. Thus, leaving the “clean” data for last could allow the model to better exploit the easy data. We will add this possibility to the revision. Even without a clear intuition, our contribution here is to show, in an identical setting and without finite size effects, that both anticurriculum and curriculum can indeed outperform the baseline."

**Authors' response to reviewer S1ip:**

1- "Overall, the analytical solution is between 2 and 6 orders of magnitude faster. We will add this approximate speed-up factor and discussion to the supplement (or if space allows, the revision)."

2- "The large input limit means that the input size N and the dataset size M go to infinity with finite ratio \alpha=M/N. This is an important point that must have gotten lost during the iterations, thank you for catching this. We will reintroduce it in the revised version."

3- "Figure 1 is based on notations and definitions introduced in section 3 and is not easily understood at the point it is presented. Reply: We will replace the image with explicit notation."

4- "Line 143: "starting from a large initialisation", I assume it means the random initialization scale of the student or teacher network. It is not entirely clear. Please include more details about this initialization. Does it use a normal random distribution? Reply: Yes, we use a normal distribution with a fixed variance. When we refer to large/small initialization we mean large/small initial variance. We will clarify this in the revised version."

5- "Figure 1: "The curriculum boundary lies at α = 1/2". What is the curriculum boundary? It is never defined. The abstract talks about "curriculum boundary consolidation", but it is not further elaborated. Reply: We mean the switching point between the two levels of difficulty. We will clarify this."

6- "Half-way through the paper section numbers stop being used. What I assume are sections 5 and 6 do not have numbers. And I am not sure if the subsections of Section 4 belong there. Reply: We will add 5 and 6 to the last two sections."

7- "Figure 3c "Accuracy hard samples." is not an accurate title. It is accuracy of all samples, easy and hard. It's just that anti-curriculum performs best, which is influenced by hard samples. Reply: We will replace it with just accuracy."

8- "The figure captions are inconsistent in the uses of (a), (b), (left, center, right), and (top, bottom). For example Figure 3 says (a), (b) and then "the right panel". Reply: We will use only the a,b,c notation."

9- "As stated in the paper, the answer does depend on the setup and in principle we would expect different behaviours for non-convex models. In particular, the presence of different basins of attraction towards different minima would suggest that initializing close to a good one would produce a performance improvement. However, note that empirical results in the ML field are not showing clear signals in this direction. A possible explanation is that relying on memory effects in the learning dynamics would require one to hit a sweet spot in the learning rate value and in the number of training epochs, and this seems hard to be achieved consistently. For this reason, we speculate that explicitly enforcing this memory by altering the loss with the curriculum information could be useful even in these settings. We will further emphasize that this statement about memorylessness applies only to our setting."

10- "We thank the reviewer for pointing out several elements that were missing in this version: we will make sure to include them in the revised version of this paper."

**Award:**

No

---

### Decision · Program_Chairs · 2022-09-14

Accept